# Direct Preference Optimization for Primitive-Enabled Hierarchical RL: A Bilevel Approach

**Utsav Singh**[1*]**, Souradip Chakraborty**[2]**, Wesley A. Suttle**[3]**, Brian M. Sadler**[4]**,
Derrik E. Asher,**[3] **Anit Kumar Sahu,**[5] **Mubarak Shah,**[6] **Vinay P. Namboodiri,**[7]
**Amrit Singh Bedi**[6]

[1]IIT Kanpur, India, [2]University of Maryland, College Park, MD, USA
[3]U.S. Army Research Laboratory, Adelphi, MD, USA, [4]University of Texas, Austin, TX, USA
[5]Oracle, [6]University of Central Florida, Orlando, FL, USA, [7]University of Bath, Bath, UK
[*]Work done as a visiting research scholar at the University of Central Florida
utsavz@iitk.ac.in

## Abstract

Hierarchical reinforcement learning (HRL) enables agents to solve complex, long-horizon tasks by decomposing them into manageable sub-tasks. However, HRL methods face two fundamental challenges: (i) non-stationarity caused by the evolving lower-level policy during training, which destabilizes higher-level learning, and (ii) the generation of infeasible subgoals that lower-level policies cannot achieve. To address these challenges, we introduce DIPPER, a novel HRL framework that formulates goal-conditioned HRL as a bi-level optimization problem and leverages direct preference optimization (DPO) to train the higher-level policy. By learning from stationary preference comparisons over subgoal sequences rather than rewards that depend on the evolving lower-level policy, DIPPER mitigates the impact of non-stationarity on hierarchical learning. To address infeasible subgoals, DIPPER incorporates lower-level value function regularization that encourages the higher-level policy to propose achievable subgoals. We also introduce two novel metrics to quantitatively verify that DIPPER mitigates non-stationarity and infeasible subgoal generation issues in HRL. We perform empirical evaluations on challenging robotic navigation and manipulation benchmarks and show that DIPPER achieves upto $40\%$ improvements over state-of-the-art baselines, demonstrating that preference-based methods can effectively alleviate persistent challenges in hierarchical learning.

## 1 Introduction

Hierarchical Reinforcement Learning (HRL) offers an effective framework for tackling complex, long-horizon tasks by decomposing them into manageable sub-tasks (Sutton et al., 1999; Harb et al., 2018). In goal-conditioned HRL (Dayan & Hinton, 1992; Vezhnevets et al., 2017), a higher-level policy sets subgoals for a lower-level policy (henceforth called the *primitive policy*), which executes lower level primitive actions to achieve these subgoals. This decomposition enables temporal abstraction and improves exploration efficiency (Nachum et al., 2019).

**Challenges.** However, HRL methods face two persistent challenges, especially in sparse reward settings: *(i) Non-stationarity-* The higher-level policy's learning process becomes unstable due to the evolving nature of the lower-level policy (Levy et al., 2019; Nachum et al., 2018). As the lower-level policy updates, the higher-level reward function and transition dynamics shift, leading to a non-stationary environment that hinders learning. *(ii) Infeasible subgoal generation-* The higher-level policy might generate subgoals that are beyond the current capabilities of the lower-level policy, resulting in suboptimal performance (Chane-Sane et al., 2021). These challenges stem from the intertwined dependencies between the hierarchical levels. The higher-level policy's rewards and transitions depend on the lower-level policy's behavior, while subgoals from the higher level simultaneously shape the lower-level policy's actions. This bidirectional dependency creates a complex optimization landscape that traditional HRL approaches fail to solve.

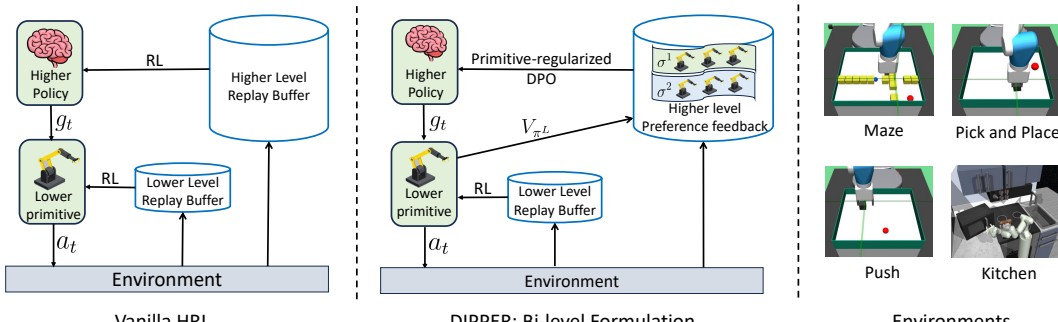

Figure 1: **DIPPER Overview: (left)** In vanilla HRL, the higher level predicts subgoals $g_t$ and gets the environment reward that depend on the lower primitive behavior, which causes non-stationarity in HRL. Also, the higher level may predict infeasible subgoals that are too hard for lower primitive. **(middle)** In `DIPPER`, the lower level value function $V_{\pi^L}$ is leveraged to condition higher level policy into predicting feasible subgoals, and direct preference optimization (`DPO`) is used to optimize higher level policy. Since this preference-based learning approach does not depend on lower primitive, this mitigates non-stationarity. Note that since the *current* estimation of value function is used to regularize the higher policy, it does not cause non-stationarity. **(right)** Training environments: ($i$) maze navigation, ($ii$) pick and place, ($iii$) push, and ($iv$) franka kitchen.

**Lack of a principled formulation.** We posit that a key reason for these persistent challenges is the absence of a mathematically precise formulation that fully captures the inter-dependencies between hierarchical policies. To address this, we model `HRL` as a bi-level optimization problem, where the higher-level policy optimization constitutes the upper-level problem and the lower-level policy optimization forms the lower-level problem. This unified framework enables the joint optimization of both policies while explicitly modeling their inter-dependencies. Thus, we propose a bi-level reformulation tailored to goal-conditioned `HRL` (Section 4.1).

**A bi-level view clarifies the true sources of instability.** While our bi-level formulation models dependencies between hierarchical levels, fundamental challenges still remain: the higher-level policy's reward depends on the evolving lower-level policy, resulting in an unstable and non-stationary learning signal. To address these limitations, we introduce `DIPPER`, a `HRL` method that leverages direct preference optimization (DPO) (Rafailov et al., 2023) to train the higher-level policy on preference datasets where trajectory pairs are ranked by stationary preference labels that, unlike environment rewards, do not depend on changing lower-level policy behavior. By optimizing the higher-level policy with DPO on this stationary dataset, we decouple higher-level learning from the non-stationary lower-level policy behavior, thus mitigating non-stationarity due to higher-level rewards and stabilizing hierarchical training. Further, our bi-level formulation enables us to address the infeasible subgoal generation problem at the higher level, by incorporating lower-level value function regularization; this is a direct consequence of our bi-level formulation and grounds subgoal proposals in the lower level's value function, ensuring that the higher-level policy generates feasible subgoals.

Our main contributions are as follows:

1. **Bi-level optimization framework for `HRL`:** We provide a mathematical formulation of `HRL` as a bi-level optimization problem, capturing the interdependencies between hierarchical policies and enabling principled algorithmic development (Section 4).

2. **Mitigation of non-stationarity and infeasible subgoal generation:** By adopting a bi-level approach, `DIPPER` leverages `DPO` to reduce the effects of non-stationarity and infeasible subgoal generation, as demonstrated through ablation studies, analysis and novel metrics (Section 5 Figure 3).

3. **Improved performance in complex robotics tasks:** Extensive experiments across diverse navigation and manipulation environments show that `DIPPER` achieves upto 40% improvement over state-of-the-art baselines in complex pick and place, push and franka kitchen tasks where other methods fail to make any meaningful progress (Section 5).

## 2 RELATED WORK

**Hierarchical Reinforcement Learning (`HRL`).** `HRL` offers the benefits of temporal abstraction and improved exploration (Nachum et al., 2019), enabling agents to solve complex, long-horizon tasks by decomposing them into sub-tasks (Sutton et al., 1999; Barto & Mahadevan, 2003; Parr & Russell, 1998). Despite these advantages, `HRL` methods are fundamentally challenged by non-stationarity (Nachum et al., 2018; Levy et al., 2019) and the generation of infeasible subgoals. Prior works have sought to mitigate non-stationarity by simulating optimal lower-level primitive behavior (Levy et al., 2019), relabeling transitions in the replay buffer (Nachum et al., 2018; Singh & Namboodiri, 2023b;a), or assuming access to privileged information such as expert demonstrations or preferences (Singh et al., 2024; Singh & Namboodiri, 2023a;b). However, existing methods lack a principled mathematical framework to explicitly model the bidirectional dependencies between hierarchical policies, where the higher-level policy's subgoal predictions influence lower-level behavior, while the evolving lower-level policy simultaneously destabilizes higher-level learning. To address this, we reformulate `HRL` as a bi-level optimization problem, explicitly decoupling and coordinating the interdependent objectives of hierarchical policies through mathematical constraints.

**Preference-based Learning (`PbL`).** `PbL` applies reinforcement learning to human preference data (Knox & Stone, 2009; Pilarski et al., 2011; Wilson et al., 2012; Daniel et al., 2015), providing a mechanism for guiding policy learning in the absence of explicit reward signals. Prior approaches (Christiano et al., 2017; Lee et al., 2021) first train a reward model from human preferences and then optimize a policy based on this reward. Prior work (Singh et al., 2024) attempts to address `HRL` non-stationarity by leveraging reinforcement learning from human feedback (RLHF) (Christiano et al., 2017), by learning a reward function for the higher-level policy, avoiding direct dependence on the non-stationary environment rewards.

More recently, direct preference optimization (`DPO`) methods (Rafailov et al., 2023; 2024; Hejna et al., 2023) have emerged, which directly optimize the policy using a KL-regularized maximum likelihood objective, bypassing the need for an explicit reward model. Our work builds on advances in maximum entropy RL (Ziebart, 2010) and `DPO`, deriving a `DPO` objective regularized by the lower-level policy's value function to address both non-stationarity and infeasible subgoal generation issues in `HRL`. For a comprehensive review of related work, see Appendix A.3.

## 3 PROBLEM FORMULATION

### 3.1 HIERARCHICAL REINFORCEMENT LEARNING (HRL)

**Hierarchical Setup:** The hierarchical formulation consists of two levels: a higher-level policy and a lower-level policy. Let $L$ represent the overall task horizon, which is factorized as $L = T \times K$, where $T$ and $K$ denote the horizons of the higher-level and lower-level policies, respectively. The higher-level policy generates subgoals every $K$ timesteps, while the lower-level policy executes primitive actions to achieve these subgoals within the $K$-timestep window. Let $t \in [1, T]$ and $k \in [1, K]$ denote the timesteps for the higher-level and lower-level policies, respectively. We denote the timestep indexes for higher and lower levels separately.

**Lower Level MDP:** The lower level MDP is defined as $(\mathcal{S}, \mathcal{A}^L, p^L, r^L)$, where $\mathcal{S}$ is the state space, and $p^L : \mathcal{S} \times \mathcal{A}^L \to \Delta(\mathcal{S})$ denotes the transition dynamics. The lower level action space is denoted as $\mathcal{A}^L$. The lower-level policy $\pi^L : \mathcal{S} \times \mathcal{G} \to \Delta(\mathcal{A}^L)$ generates primitive actions $a_k \sim \pi^L(\cdot|s_{t+k}, g_t)$ conditioned on subgoals $g_t \in \mathcal{A}^H$ provided by the higher-level policy, and $s_{t+k} \in \mathcal{S}$ is the current state. The lower-level policy is sparsely rewarded when it achieves the subgoal $g_t$: $r^L(s_{t+k}, a_k, g_t) = \mathbf{1}_{\{|s_{t+k} - g_t|^2 < \varepsilon\}}$, with $\mathbf{1}_C$ as an indicator function returning 1 if the condition $C$ holds, indicating that the subgoal $g_t$ is achieved. In the lower-level replay buffer, a transition is defined as $(s_{t+k}, g_t, a_k, r^L(s_{t+k}, a_k, g_t), s_{t+k+1})$. We adopt a maximum entropy RL setting, where $\mathcal{H}(\pi^L)$ denotes the entropy of the lower-level policy $\pi^L$. To learn optimal lower level policy, we maximize the expected lower level cumulative reward, formally defined as $\pi_*^L := \arg\max_{\pi^L} V^L(\pi^H)$, where

$$V^L(\pi^H) = \mathbb{E}_{g_t \sim \pi^H, a_k \sim \pi^L} \left[ \sum_{k=0}^{K-1} r^L(s_{t+k}, a_k, g_t) + \lambda \mathcal{H}(\pi^L) \right]. \tag{1}$$

Here $g_t \sim \pi^H(\cdot|s_t, g^*)$ is the subgoal selected by the upper level for step $t$, and expectation is over the randomness induced by the environment transitions, lower level policy, and higher level policy. For each step $t$, we have $a_k \sim \pi^L(\cdot|s_{t+k}, g_t)$, and for the current state $s_{t+k}$, the next state is $s_{t+k+1} \sim p^L(s_{t+k}, a_k, g_t)$ where $p^L$ determines the environment dynamics, and is hence stationary.

**Higher-Level MDP:** The higher-level MDP is defined as $(\mathcal{S}, \mathcal{G}, \mathcal{A}^H, p^H, r^H)$, where $\mathcal{S}$ is the state space, and $\mathcal{G}$ is the goal space. $\mathcal{A}^H$ is the higher-level action space, which we set as $\mathcal{A}^H = \mathcal{G}$ (subgoals are drawn from the goal space). The environment reward for the higher level, $r^H : \mathcal{S} \times \mathcal{A}^H \times \mathcal{G} \to \mathbb{R}$, encourages progress toward the final goal $g^*$. A higher-level transition is stored in the replay buffer as $(s_t, g^*, g_t, r^H(s_t, g_t, g^*), s_{t+1})$. We adopt a maximum entropy RL setting, where $\mathcal{H}(\pi^H)$ denotes the entropy of the higher-level policy $\pi^H$. The higher-level objective is $\pi_*^H := \arg\max_{\pi^H} J_H(\pi^H, \pi^L(\pi^H))$, where

$$J_H(\pi^H, \pi^L(\pi^H)) := \mathbb{E}\left[\sum_{t=0}^{T-1} r(s_t, g_t, g^*) + \lambda \mathcal{H}(\pi^H)\right]. \tag{2}$$

Here, $\lambda$ is the entropy weight-parameter. This objective can be expressed using the higher-level value function $V^H(s_t, g^*; \pi^H, \pi^L)$, which estimates the expected cumulative reward starting from state $s_t$ toward goal $g^*$, following policy $\pi^H$ at the higher level and $\pi^L$ at the lower level. The higher-level Q-function $Q^H(s_t, g^*, g_t; \pi^H, \pi^L)$ estimates the expected cumulative reward for taking subgoal $g_t$ in state $s_t$ to achieve the final goal $g^*$.

**Non-stationarity in HRL.** For every subgoal $g_t \sim \pi^H(\cdot|s_t, g^*)$ predicted by the higher-level policy, the lower level policy $\pi^L$ is allowed to execute for $k$ timesteps, after which the policy reaches state $s_{t+k}$. The transition kernel $p_{\pi^L}^H(s_t, g_t)$ models the distribution over next states $s_{t+k} \sim p_{\pi^L}^H(s_t, g_t)$ depends explicitly on the lower-level policy $\pi^L$ (as the lower-level policy determines the sequence of primitive actions executed to pursue $g_t$ over $k$ steps). Since $\pi^L$ is updated throughout training, both the higher-level reward $r^H$ (which depends on the achieved state $s_{t+k}$) and transition kernel $p_{\pi^L}^H$ become non-stationary. This causes training instability in learning $\pi^H$. Further, in off-policy RL, since the lower level policy changes during training, prior transitions in the replay buffer become obsolete, further exacerbating the non-stationarity issue.

We now enlist the challenges of standard `HRL`-based approaches.

**Challenges of `HRL`:**

While `HRL` offers advantages over `RL`, such as higher sample efficiency than flat RL through temporal abstraction and enhanced exploration (Nachum et al., 2019), it faces two fundamental challenges:

**C1: Non-stationarity.** Vanilla off-policy `HRL` suffers from non-stationarity due to changing behavior of the lower-level policy (Nachum et al., 2018; Levy et al., 2019), due to which the higher level reward function and transition dynamics become non-stationary, thus causing HRL non-stationarity.

**C2: Infeasible subgoal generation.** Since the sub-optimality in the lower-level policy affects its ability to reach a given subgoal, it consequently impacts the higher-level credit assignment during subgoal generation. This causes the higher level to produce infeasible subgoals for the lower level policy (Chane-Sane et al., 2021). Thus, despite its theoretical advantages, `HRL` often underperforms in practice (Nachum et al., 2018).

Given these challenges, preference-based learning (PbL) methods emerge as a promising alternative by incorporating stationary human feedback to guide policy optimization without direct dependence on shifting rewards or transitions. PbL approaches, such as Reinforcement Learning from Human Feedback (RLHF) (Christiano et al., 2017) and Direct Preference Optimization (DPO) (Rafailov et al., 2023), rank trajectories via pairwise preferences to train models in complex, reward-sparse tasks. In what follows, we outline PbL fundamentals and the limitations of directly applying PbL approaches.

### 3.2 Preference Based Learning (PbL)

Preference-based learning (PbL) methods such as `RLHF` (Christiano et al., 2017; Ibarz et al., 2018; Lee et al., 2021) and `DPO` (Rafailov et al., 2023) leverage preference data to solve complex tasks.

**RL from human feedback (`RLHF`):** In this setting, the agent behavior is represented as a $T$-length trajectory denoted as $\tau$ of states and actions: $\tau = ((s_t, g_t), (s_{t+1}, g_{t+1})...(s_{t+T-1}, g_{t+T-1}))$. The

learned reward model to be learned is denoted by $r : \mathcal{S} \times \mathcal{G} \to \mathbb{R}$, with parameters $\phi$. The preferences between two trajectories, $\tau^1$ and $\tau^2$, can be expressed through the Bradley-Terry model (Bradley & Terry, 1952) $P_\phi \left[ \tau^1 \succ \tau^2 \right] = \frac{\exp \sum_t r\left(s_t^1, g_t^1, g^*\right)}{\sum_{i \in \{1,2\}} \exp \sum_t r\left(s_t^i, g_t^i, g^*\right)}$, where $\tau^1 \succ \tau^2$ implies that $\tau^1$ is preferred over $\tau^2$. The preference dataset $\mathcal{D}$ has entries of the form $(\tau^1, \tau^2, y)$, where $y = (1, 0)$ when $\tau^1$ is preferred over $\tau^2$, $y = (0, 1)$ when $\tau^2$ is preferred over $\tau^1$, and $y = (0.5, 0.5)$ in case of no preference. In RLHF, we first learn the reward function $r$ (Christiano et al., 2017) using cross-entropy loss along with Bradley-Terry model to yield the formulation:

$$\mathcal{L} = -\mathbb{E}\left[\log \sigma\left(\sum_{t=0}^{T-1} r\left(s_t^1, g_t^1, g^*\right) - \sum_{t=0}^{T-1} r\left(s_t^2, g_t^2, g^*\right)\right)\right], \tag{3}$$

where expectation is over $(\tau^1, \tau^2, y) \sim \mathbb{D}$.

**Direct Preference Optimization (`DPO`):** Although RLHF provides a framework for learning policies from preferences, it involves RL training step which is often computationally expensive and unstable in practice. In contrast, DPO (Rafailov et al., 2023) circumvents the need for RL step by using a closed-form solution for the optimal policy of the KL-regularized RL problem (Levine, 2018): $\pi^*(a|s) = \frac{1}{Z(s)} \pi_{ref}(a|s) e^{r(s,a)}$, where $\pi_{ref}$ is the reference policy, $\pi^*$ is the optimal policy, and $Z(s)$ is a normalizing partition function. DPO directly optimizes policies from preferences, bypassing explicit reward modeling, offering stable convergence (detailed in Section 5).

**Challenges of directly applying `PbL` to `HRL`:**

**Directly using `RLHF`:** Prior approaches leverage the advancements in PbL to mitigate HRL non-stationarity (Singh et al., 2024) by utilizing the reward model $r_\phi^H$ learned using the reward model (corresponding to the preference dataset) as higher level rewards instead of environment rewards $r_{\pi_L}^H$ used in vanilla HRL approaches, which depend on the sub-optimal lower primitive. However, such approaches may lead to degenerate solutions by generating infeasible subgoals for the lower-level primitive. Additionally, such approaches require RL as an intermediate step, which might cause training instability (Rafailov et al., 2023).

**Directly using `DPO`:** In temporally extended task environments like robotics, directly extending DPO to the HRL framework is non-trivial due to three reasons: $(i)$ such scenarios deal with multi-step trajectories involving stochastic transition models, $(ii)$ efficient pre-trained reference policies are typically unavailable in robotics, $(iii)$ similar to RLHF, such approaches may produce degenerate solutions when higher level policy subgoal predictions are infeasible.

## 4 Proposed Approach

To address the dual challenges of non-stationarity (C1) and infeasible subgoal generation (C2) in HRL, we introduce DIPPER: **DI**rect **P**reference Optimization for **P**rimitive-**E**nabled Hierarchical **R**einforcement Learning. We first formulate HRL as a bi-level optimization problem to develop a principled framework that fully captures the bi-directional dependence between hierarchical policies (Section 4.1). Subsequently, we introduce DIPPER, which decouples higher-level policy optimization from the non-stationary lower-level policy behavior by leveraging DPO (Rafailov et al., 2023) to train the higher level policy and RL to train the lower-level policy, thus mitigating HRL non-stationarity (**C1**) (Section 4.2). Additionally, our principled bi-level formulation naturally yields a lower-level value function based regularization which ensures that the generated subgoals remain feasible for the lower-level policy, thus addressing the infeasible subgoal generation issue (**C2**) in HRL. Finally, we derive DIPPER objective, analyze its gradient, and provide the final practical algorithm.

### 4.1 HRL: Bi-Level Formulation

We present our bi-level formulation by using equation 2 and representing it as a constrained optimization problem assuming the lower level policy to be optimal:

$$\max_{\pi^H, \pi^L} \mathcal{J}(\pi^H, \pi_*^L(\pi^H)) \quad s.t. \quad \pi_*^L(\pi^H) = \arg\max_{\pi^L} V^L(\pi^H), \tag{4}$$

where $\mathcal{J}(\pi^H, \pi_*^L(\pi^H))$ is the higher level objective and $V^L(\pi^H)$ is the lower level value function, conditioned on higher level policy subgoals. Utilizing the recent advancements in the optimization literature (Liu et al., 2022), we represent equation 4 by equivalent constrained optimization problem:

$$\max_{\pi^H, \pi^L} \mathcal{J}(\pi^H, \pi^L) \quad s.t. \quad V^L(\pi^H) - V_*^L(\pi^H) \geq 0. \tag{5}$$

where, $V_*^L(\pi^H) = \max_{\pi_L} V^L(\pi^H)$. Notably, since the left-hand side of the inequality constraint is always non-positive due to the fact that $V^L(\pi^H) - V_*^L(\pi^H) \leq 0$, the constraint is satisfied only when $V^L(\pi^H) = V_*^L(\pi^H)$. Although the constraint in equation 5 holds for all states $s$ and subgoals $g$, however, enforcing it globally would make the problem intractable. Therefore, we relax the constraint by only considering the $(s_t, g_t)$ pairs traversed by the higher-level policy. Leveraging this relaxed constraint and replacing $\mathcal{J}(\pi^H, \pi^L)$ from equation 2, we propose the following approximate Lagrangian objective

$$\max_{\pi^H, \pi^L} \mathbb{E}\left[\sum_{t=0}^{T-1}(r(s_t, g_t, g^*) + \lambda\mathcal{H}(\pi^H) + \lambda(V^L(s_t, g_t) - V_*^L(s_t, g_t)))\right]. \tag{6}$$

We can use equation 6 to solve the `HRL` policies for both higher and lower level, where $(i)$ the higher level policy learns to achieve the final goal and predict feasible subgoals to the lower level policy, and $(ii)$ the lower level policy learns to achieve the predicted subgoals. However, directly optimizing equation 6 requires knowledge of higher level reward function $r_{\pi_L}^H$ which is a function of $\pi^L$, which is non-stationary. Further, using RL to optimize the objective may lead to unstable learning (Rafailov et al., 2023).

## 4.2 DIPPER

To overcome these challenges, we propose `DIPPER`, our hierarchical approach that leverages primitive-regularized `DPO` objective to optimize the hierarchical policies. Using the objective in equation 6 to derive the reward-policy equivalence equation and replacing it in equation 3, we get the following `DIPPER` objective:

$$\mathcal{L}_{\mathcal{O}} = -\mathbb{E}_{(\tau^1, \tau^2, y)\sim\mathbb{D}}\left[\log\sigma\left(\sum_{t=0}^{T-1}\left(\beta\log\pi_*^H\left(g_t^1|s_t^1, g^*\right) - \beta\log\pi_*^H\left(g_t^2|s_t^2, g^*\right)\right.\right.\right.$$

$$\left.\left.\left. -\lambda((V^L(s_t^1, g_t^1) - V_*^L(s_t^1, g_t^1)) - (V^L(s_t^2, g_t^2) - V_*^L(s_t^2, g_t^2))))\right)\right)\right]. \tag{7}$$

where $(\tau^1, \tau^2, y) \sim \mathbb{D}$ represents a preference pair sampled from the preference dataset $\mathbb{D}$, $\sigma$ represents the sigmoid function, $\lambda$ is regularization weight parameter, and $\beta$ is entropy weight parameter. The complete derivation of the DIPPER objective is provided in the Appendix A.1. This `DIPPER` objective optimizes the higher-level policy using primitive regularized `DPO`, thus decoupling the higher-level learning from non-stationary lower-level policy rewards.

**Analyzing `DIPPER` gradient:** We further analyze the `DIPPER` objective by interpreting the gradient with respect to higher level policy $\pi_*^H$, denoted as:

$$\nabla\mathcal{L}_{\mathcal{O}} = -\beta\mathbb{E}_{(\tau_1, \tau_2, y)\sim\mathbb{D}}\left[\sum_{t=0}^{T-1}\left(\underbrace{\sigma\left(\hat{r}\left(s_t^2, g_t^2\right) - \hat{r}\left(s_t^1, g_t^1\right)\right)}_{\text{higher weight for incorrect preference}}\right.\right.$$

$$\left.\left.\cdot\left(\underbrace{\nabla\log\pi^H\left(g_t^1|s_t^1, g^*\right)}_{\text{increase likelihood of }\tau_1} - \underbrace{\nabla\log\pi^H\left(g_t^2|s_t^2, g^*\right)}_{\text{decrease likelihood of }\tau_2}\right)\right)\right]. \tag{8}$$

where $\hat{r}(s_t, g_t, g^*) = \beta\log\pi^H(g_t|s_t, g^*) - \lambda(V^L(s_t, g_t) - V_*^L(s_t, g_t))$, which acts as an implicit reward model determined by the higher-level policy and the lower-level value function. This objective increases the likelihood of preferred trajectories while decreasing the likelihood of dis-preferred ones. Based on the strength of the KL constraint, the examples are weighted based on how inaccurately the implicit reward model $\hat{r}(s_t, g_t, g^*)$ ranks the trajectories. This implicit reward acts as a value function regularizer that conditions the higher-level policy to generate feasible subgoals.

**Practical algorithm:** The `DIPPER` objective in Eqn. 7 requires calculation of optimal lower-level value function $V_*^L(s_t, g_t)$, which is computationally expensive. We accordingly consider an approximation $V_m^L(s_t, g_t)$ to replace $V_*^L(s_t, g_t)$, where we update the value function $V_m^L(s_t, g_t)$ gradient $m$ times for every policy update, to get the following objective:

$$\mathcal{L}_\mathcal{O} = -\mathbb{E}_{(\tau^1, \tau^2, y) \sim \mathbb{D}} \bigg[ \log \sigma \bigg( \sum_{t=0}^{T-1} \big( \beta \log \pi_*^H (g_t^1 | s_t^1, g^*) - \beta \log \pi_*^H (g_t^2 | s_t^2, g^*) - \tag{9}$$

$$\lambda((V^L(s_t^1, g_t^1) - V_m^L(s_t^1, g_t^1) - (V^L(s_t^2, g_t^2) - V_m^L(s_t^2, g_t^2)))) \bigg) \bigg].$$

### 4.3 HIGHER-LEVEL PREFERENCE DATASET COLLECTION

In the vanilla PbL framework, preferences are elicited from human feedback (Christiano et al., 2017). However, collecting large amounts of human preference feedback for PbL is computationally expensive. To avoid this, we follow the primitive-in-the-loop (PiL) approach in PIPER (Singh et al., 2024) and collect preference feedback using implicit sparse rewards $\hat{r}(s_t, g^*, g_t)$ to determine labels $y$ between trajectories $\tau^1$ and $\tau^2$. Let us assume that a higher-level trajectory from the preference dataset is represented as $\tau = \{(s_{t*k}, g_t)\}_{t=0}^{n-1}$. For a collected trajectory $\tau = \{(s_{t*k}, g_t)\}_{t=0}^{n-1}$ and final goal $g^*$, we compute per-step reward $\hat{r}(s_{t*k}, g^*, g_t) = \mathbf{1}_{\{\|s_{t*(k+1)} - g^*\|_2 \leq \epsilon\}}$ where $s_{t*(k+1)}$ is the fixed state reached after $k$ timesteps toward $g_t$ from $s_{t*k}$. Notably, this per-step reward $\hat{r}$ depends on the reached states $s_{t*(k+1)}$ and not on the lower level policy behavior. The total score for a trajectory $\tau$ is computed as $\hat{R}(\tau) = \sum_{t=0}^{n-1} \hat{r}(s_{t*k}, g^*, g_t)$. For pairs $(\tau^1, \tau^2)$, label $y$ prefers $\tau^1 \succ \tau^2$ if its total score $\hat{R}(\tau^1)$ exceeds $\hat{R}(\tau^2)$. Since the preference labels of these trajectory pairs do not depend on the lower-level policy behavior, the collected preference dataset remains stationary. This stationary preference dataset is used to train the higher-level policy via DPO, thus mitigating HRL non-stationarity. The pseudo-code for `DIPPER` is provided in the Appendix A.2.

## 5 EXPERIMENTS

In our empirical analysis, we ask the following questions:

**(1) How well does `DIPPER` perform against baselines?** How well does `DIPPER` perform in complex robotics control tasks against prior hierarchical and non-hierarchical baselines?

**(2) Does `DIPPER` mitigate HRL limitations?** How well does `DIPPER` mitigate the issues of non-stationarity **(C1)** and infeasible subgoal generation **(C2)** in `HRL`?

**(3) What is the impact of our design decisions on the overall performance?** Can we concretely justify our design choices through extensive ablation analysis?

**Task details**: We assess `DIPPER` on four robotic navigation and manipulation environments: $(i)$ maze navigation, $(ii)$ pick and place (Andrychowicz et al., 2017), $(iii)$ push, and $(iv)$ franka kitchen environment (Gupta et al., 2019). These are formulated as sparse reward scenarios, where the agent is only rewarded when it comes within a $\delta$ distance of the goal. Due to this, these environments are hard where the agent must extensively explore the environment before coming across any rewards. As an example: in franka kitchen task, the agent only receives a sparse reward after achieving the final goal (e.g. successfully opening the microwave and then turning on the gas knob).

**Environment details**: We provide the implementation and environment details in Appendix A.6 and A.7, and the implementation code in the supplementary. We provide the preference data analysis in Appendix A.4 The main objective of our empirical analysis is to evaluate our approach on sparse-reward long-horizon tasks. In the franka kitchen task, the agent is sparsely rewarded only when it completes the final task, e.g open the microwave and turn on the gas knob. These nuances prohibit prior baselines from performing well in these tasks, which makes these test beds appropriate for empirical evaluations. Finally, for harder tasks such as pick and place, push and franka kitchen, we assume access to one human demonstration and incorporate an imitation learning objective at the lower level to accelerate learning. However, we apply the same assumption consistently across all baselines to ensure fairness.

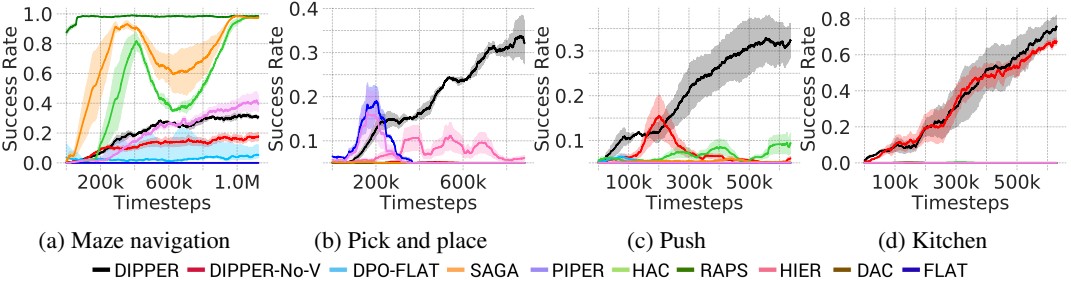

Figure 2: **Success Rate plots.** This figure illustrates the success rates across four sparse-reward maze navigation and robotic manipulation tasks, where the solid lines represent the mean, and the shaded areas denote the standard deviation across 5 different seeds. We evaluate DIPPER against several baselines. Although HAC, SAGA and RAPS outperform DIPPER in the easier maze task, they fail to perform well in other challenging tasks, where DIPPER demonstrates strong performance and significantly outperforms the baselines.

We employ DPO instead over standard policy gradient methods due to stable convergence properties without explicit reward modeling or intermediate RL steps (detailed comparisons against policy gradient methods are provided in Experiment section 5). We tune the hyper-parameters via grid search, with ablation studies showing balanced values yield optimal performance without extreme sensitivity.

### (1) How well does `DIPPER` perform against baselines?

In this section, we compare DIPPER against multiple hierarchical and non-hierarchical baselines. Please refer to Figure 2 for success rate comparison plots and subsequent discussion. The solid line and shaded regions represent the mean and standard deviation, averaged over 5 seeds.

**Comparison with `DPO` Baselines.**

**DIPPER-No-V baseline:** This is an ablation of DIPPER without primitive regularization. The primitive regularization approach in DIPPER regularizes the higher level policy to predict feasible subgoals. We employ this baseline to highlight the critical role of generating feasible subgoals. As shown in Figure 2, DIPPER outperforms this baseline, demonstrating the importance of feasible subgoal generation in achieving superior performance.

**DPO-FLAT baseline:** This is a single-level DPO (Rafailov et al., 2024) implementation. Note that since we do not have access to a pre-trained model as a reference policy in robotics scenarios like generative language modeling, we use a uniform policy as a reference policy, which effectively translates to an additional objective of maximizing the entropy of the learnt policy. DIPPER is an hierarchical approach which benefits from temporal abstraction and improved exploration, as seen in Figure 2 which shows that DIPPER significantly outperforms this baseline.

**Comparison with Hierarchical Baselines.**

**SAGA baseline:** We compare DIPPER with SAGA (Wang et al., 2023), a hierarchical approach that employs state conditioned discriminator network training to address non-stationarity, by ensuring that the high-level generates subgoals that align with the current state of the low-level policy. We find that although SAGA performs well in the maze task, it fails to solve harder tasks where DIPPER significantly outperforms. This demonstrates that SAGA suffers from non-stationarity in harder long horizon tasks, whereas DIPPER is able to better mitigate non-stationarity issue in such tasks.

**PIPER baseline:** This baseline (Singh et al., 2024) leverages RLHF to learn higher level reward function to address HRL non-stationarity. To ensure fair comparison, we implement an ablation of PIPER without HER (Andrychowicz et al., 2017). DIPPER is able to outperform this PIPER ablation on all tasks, showing that our DPO based approach avoids training instability caused by RL, and is able to better mitigate non-stationarity in HRL.

**RAPS baseline:** Here, we consider RAPS (Dalal et al., 2021) baseline, which employs behavior priors at the lower level for solving the task. Although RAPS is a framework for solving robotic tasks where behavior priors are readily available, it requires considerable effort to construct such priors and struggles to perform well in their absence, especially when dealing with sparse reward scenarios.

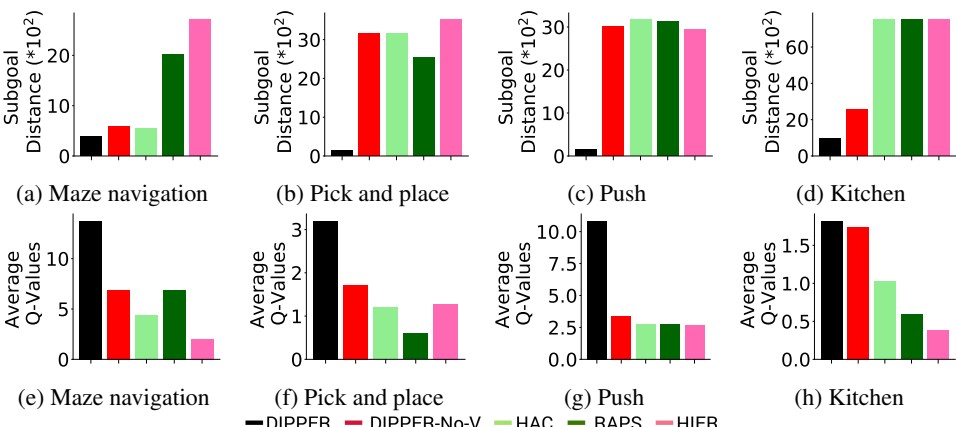

Figure 3: **(Row 1) Subgoal Distance Metric.** We compare `DIPPER` with `DIPPER-No-V`, `HAC`, `RAPS`, `HIER` baselines, based on average distance between subgoals predicted by the higher level policy and subgoals achieved by the lower level primitive. `DIPPER` consistently generates low average distance values, which implies that in `DIPPER`, the higher level policy generates achievable subgoals that induce optimal lower primitive goal reaching behavior. This shows that DIPPER is able to address non-stationary in HRL and generate feasible subgoals. **(Row 2) Lower Q-Function Metric.** We compare `DIPPER` with `DIPPER-No-V`, `HAC`, `RAPS`, `HIER` baselines, based on average lower level Q function values for the subgoals predicted by the higher level policy. `DIPPER` consistently produces large Q-function values, thus inducing optimal lower policy behavior thus mitigating non-stationarity and predicting feasible subgoals.

Indeed we empirically find this to be the case, since although `RAPS` performs exceptionally well in maze navigation task, it fails to perform well in other sparse complex manipulation tasks.

**HAC baseline:** We also implement `HAC` (Levy et al., 2019) baseline that mitigates non-stationarity in HRL by simulating optimal lower level primitive behavior. `HIRO` (Nachum et al., 2018) is another such baseline that addresses non-stationarity, however since `HAC` has been found to outperform `HIRO`, we chose to compare with *HAC*. Although `HAC` performs well in maze task, it struggles to perform well in harder tasks. `DIPPER` outperforms this baseline in 3 out of 4 tasks.

**HIER baseline:** We also implement `HIER`, a vanilla `HRL` baseline implemented using `SAC` (Haarnoja et al., 2018) at both hierarchical levels, but it fails to outperform `DIPPER` on any task.

**Comparison with Non-Hierarchical Baselines.**

**DAC baseline:** (`DAC`) (Kostrikov et al., 2018) is a single-level baseline using one demonstration per task. Despite having access to privileged information, `DAC` still struggles to perform well.

**FLAT baseline:** We also implement a single-level `SAC` policy, but it fails to show any progress, verifying that hierarchical abstraction is key to effective performance in complex tasks.

**(2) Does `DIPPER` mitigate HRL limitations?**

Prior work lacks principled metrics for quantifying non-stationarity **(C1)** and infeasible subgoal generation **(C2)** in `HRL`. To address this gap, we introduce two novel metrics specifically designed to measure these challenges. Using these metrics, we empirically demonstrate that `DIPPER` effectively mitigates both non-stationarity **(C1)** and infeasible subgoal generation **(C2)** in `HRL`.

**Subgoal Distance Metric.** We compare `DIPPER` with `DIPPER-No-V`, `HAC`, `RAPS`, `HIER` baselines on subgoal distance metric: the average distance between subgoals predicted by the higher level policy and subgoals achieved by the lower level primitive. A low average distance value implies that the predicted subgoals are feasible, thus inducing optimal lower level policy behavior (note that the optimal, lower-level policy is stationary, and thus avoids non-stationarity). Figure 3 (Row 1) shows that `DIPPER` consistently generates low average distance values, thus mitigating non-stationarity **(C1)**. Low average distance values imply that high-level policy in `DIPPER` generates achievable subgoals for the lower primitive due to primitive regularization, thus addressing **(C2)**.

**Lower Q-Function Metric.** Here, we compare `DIPPER` on average lower level Q function values for the subgoals predicted by the higher policy. High Q-values imply that the lower policy expects high returns for the predicted subgoals. Such subgoals are feasible and induce optimal lower primitive behavior. Figure 3 (Row 2) shows that `DIPPER` consistently leads to large Qvalues, showing that it produces subgoals with high predicted returns, indicating feasibility, and thus directly addressing both **(C1)** and **(C2)**.

### (3) What is the impact of our design choices?

We perform ablations to analyze our design choices. We first analyze the effect of varying regularization weight $\lambda$ hyper-parameter in Appendix A.5 Figure 4. $\lambda$ controls the strength of primitive regularization: if $\lambda$ is too small, we lose the benefits of primitive regularization leading to infeasible subgoals prediction. Conversely, if $\lambda$ is too large, the higher-level policy fails to achieve the final goal by repeatedly predicting trivial subgoals. We also analyze the effect of varying $\beta$ hyper-parameter in Appendix A.5 Figure 5. Excessive $\beta$ causes over-exploration, preventing optimal subgoal prediction; whereas insufficient $\beta$ limits exploration, risking suboptimal predictions.

## 6 DISCUSSION

**Limitations and future work.** Our DPO-based hierarchical formulation raises an important question: since DIPPER uses DPO for training the higher-level policy, does it generalize to out-of-distribution states and actions better than reward-model-based RL? Comparing with hierarchical RLHF could provide useful insights. Moreover, applying DIPPER to high-dimensional subgoal spaces remains challenging. We leave these directions for future work.

**Conclusion.** In this work, we introduce `DIPPER`, a novel hierarchical approach that employs primitive-regularized `DPO` to mitigate the issues of non-stationarity and infeasible generation in `HRL`. `DIPPER` employs primitive-regularized token-level `DPO` objective to efficiently learn higher level policy, and `RL` to learn the lower level primitive policy, thereby mitigating non-stationarity in `HRL`. We formulate `HRL` as a bi-level optimization objective to ensure that the higher level policy generates feasible subgoals. Based on empirical results showing up to 40% improvement, `DIPPER` is an important step towards learning effective control policies for solving complex robotics tasks.

## ETHICS STATEMENT

This work introduces DIPPER, a hierarchical reinforcement learning framework for robotic control. We acknowledge the broader societal implications of advancing autonomous robotic capabilities, particularly regarding potential impacts on automation and employment. Additionally, the training of hierarchical policies across multiple environments entails energy consumption contributing to environmental impact. However, our methodology utilizes standard robotic simulation environments and does not involve human subjects or human-annotated preference datasets, thereby eliminating risks related to annotator bias or personal data privacy.

## REPRODUCIBILITY STATEMENT

To ensure reproducibility, we provide complete mathematical formulations and proofs in Section 4.1 and Appendix A.1, with the full procedure in Algorithm 1. All hyperparameters, including learning rates and regularization coefficients ($\lambda$, $\beta$), are detailed in Appendix Table 2. We describe the configurations for all four robotic environments and baseline implementation details in Section 5 and Appendix A.6. Upon publication, we will release our complete codebase, including the bi-level optimization implementation and evaluation scripts. Computational infrastructure specifications are available in Appendix A.4 to facilitate replication.

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

CONTENTS

## A APPENDIX

### A.1 DERIVATION OF THE DIPPER OBJECTIVE

Here, we provide the complete derivation of thee `DIPPER` objective.

**1. Bi-level Objective:**

Following Eqn (7) in the main text, the approximate Lagrangian for the bi-level HRL is represented as:

$$\max_{\pi_H, \pi_L} \mathbb{E} \left[ \sum_{t=0}^{T-1} \left( r(s_t, g_t, g^*) + \lambda \mathcal{H}(\pi^H) + \lambda(V^L(s_t, g_t) - V_*^L(s_t, g_t)) \right) \right]. \tag{10}$$

This equation is a direct consequence of the proposed bi-level formulation of the HRL problem (Section 4.1). Here $r(s_t, g_t, g^*)$ is the high level reward, $\mathcal{H}(\pi^H)$ denotes the entropy with respect to the higher level policy, $V^L(s_t, g_t)$ is the lower level value function, $V_*^L(s_t, g_t)$ represents the optimal lower level value function, and $\lambda$ is the weight hyper-parameter.

Let $R_t := r(s_t, g_t, g^*) + \lambda(V^L(s_t, g_t) - V_*^L(s_t, g_t))$, then we can rewrite equation 10 as:

$$\max_{\pi^H, \pi^L} \mathbb{E} \left[ \sum_{t=0}^{T-1} R_t + \lambda \mathcal{H}(\pi^H) \right], \tag{11}$$

**2. Bellman Equation for High-Level Q-function:** Using equation 11, and as originally explored in Garg et al. (2021), the relationship between the optimal future return and the current timestep return for the higher level policy is captured by the following bellman equation:

$$Q_*^H(s_t, g^*, g_t) = \begin{cases} R_t + V_*^H(s_{t+1}, g^*) & \text{if } s_{t+1} \text{ isn't terminal,} \\ R_t & \text{if } s_{t+1} \text{ is terminal.} \end{cases} \tag{12}$$

In non-terminal cases, using the last equation we write:

$$R_t + V_*^H(s_{t+1}, g^*) = Q_*^H(s_t, g^*, g_t) \tag{13}$$

We can reformulate he items in this equation to represent the reward as:

$$R_t = Q_*^H(s_t, g^*, g_t) - V_*^H(s_{t+1}, g^*). \tag{14}$$

Thus, there is a bijection between the reward function $R_t$ and the corresponding optimal Q-function $Q_*^H(s_t, g_t, g^*)$.

**3. Trajectory-wise Expansion:**

Inspired from Rafailov et al. (2024), we consider the problem in a token-level MDP setting (for trajectory $\tau$). We consider equation 14 and take a summation over $t \in [0, T-1]$ to derive the following:

$$\sum_{t=0}^{T-1} R_t \stackrel{(a)}{=} \sum_{t=0}^{T-1} (Q_*^H(s_t, g^*, g_t) - V_*^H(s_{t+1}, g^*)) \tag{15}$$

Now, we add and substract $V_*^H(s_0, g^*)$ on the RHS:

$$\sum_{t=0}^{T-1} R_t = \sum_{t=0}^{T-1} (Q_*^H(s_t, g^*, g_t) - V_*^H(s_{t+1}, g^*)) + V_*^H(s_0, g^*) - V_*^H(s_0, g^*). \tag{16}$$

This can be re-arranged as:

$$\sum_{t=0}^{T-1} R_t = V_*^H(s_0, g^*) + \sum_{t=0}^{T-1} (Q_*^H(s_t, g^*, g_t) - V_*^H(s_t, g^*)). \tag{17}$$

Now, by definition, the advantage function $A_*^H(s_t, g^*, g_t)$ is defined as: $A_*^H(s_t, g^*, g_t) = Q_*^H(s_t, g^*, g_t) - V_*^H(s_t, g^*)$

We can re-place the term $Q_*^H(s_t, g^*, g_t) - V_*^H(s_t, g^*)$ in the last equation by $A_*^H(s_t, g^*, g_t)$ to yield:

$$\sum_{t=0}^{T-1} R_t = V_*^H(s_0, g^*) + \sum_{t=0}^{T-1} (A_*^H(s_t, g^*, g_t)). \tag{18}$$

Now, based on a result (Ziebart, 2010) for maximum entropy RL setting: $A_*^H(s_t, g^*, g_t) = \beta \log(\pi_*^H(g_t|s_t, g^*))$, we replace the advantage term in last equation to yield:

$$\sum_{t=0}^{T-1} R_t = V_*^H(s_0, g^*) + \sum_{t=0}^{T-1} (\beta \log \pi_*^H(g_t|s_t, g^*)). \tag{19}$$

Recall that we had defined $R_t = r(s_t, g_t, g^*) + \lambda(V^L(s_t, g_t) - V_*^L(s_t, g_t))$. Replacing $R_t$ on the LHS in the last equation yields:

$$\sum_{t=0}^{T-1} (r(s_t, g_t, g^*) + \lambda(V^L(s_t, g_t) - V_*^L(s_t, g_t))) = V_*^H(s_0, g^*) + \sum_{t=0}^{T-1} (\beta \log \pi_*^H(g_t|s_t, g^*)). \tag{20}$$

Re-arranging this term by bringing the $\lambda$ weighted term from LHS to RHS, we get:

$$\sum_{t=0}^{T-1} r(s_t, g_t, g^*) = V_*^H(s_0, g^*) + \sum_{t=0}^{T-1} \left( \beta \log \pi_*^H(g_t|s_t, g^*) - \lambda(V^L(s_t, g_t) - V_*^L(s_t, g_t)) \right). \tag{21}$$

## 4. Primitive-Regularized DPO Objective:

Finally, we use the LHS ($\sum_{t=0}^{T-1} r(s_t, g_t, g^*)$) derived in equation 21, and substitute it in equation 3 from the main paper, to yield our primitive regularized DPO objective $\mathcal{L}_\mathcal{O}$ as:

$$\mathcal{L}_\mathcal{O} = -\mathbb{E}_{(\tau^1, \tau^2, y) \sim \mathbb{D}} \left[ \log \sigma \left( \sum_{t=0}^{T-1} \left( \beta \log \pi_*^H(g_t^1|s_t^1, g^*) - \beta \log \pi_*^H(g_t^2|s_t^2, g^*) \right. \right. \right.$$
$$\left. \left. \left. - \lambda((V^I(s_t^1, g_t^1) - V_*^I(s_t^1, g_t^1)) - (V^I(s_t^2, g_t^2) - V_*^I(s_t^2, g_t^2))) \right) \right) \right]. \tag{22}$$

Note that terms $V_*^H(s_0, g^*)$ is the same for both trajectories and hence it cancels. This `DIPPER` objective optimizes the higher-level policy using primitive regularized `DPO`. □

### A.2 DIPPER ALGORITHM

Here, we provide the DIPPER pseudo-code.

---

**Algorithm 1** DIPPER

---

1: Initialize preference dataset $\mathcal{D} = \{\}$.
2: Initialize lower level replay buffer $\mathcal{R}^L = \{\}$.
3: **for** $i = 1 \ldots N$ **do**
4:     // Collect higher level trajectories $\tau$ using $\pi^H$ and lower level trajectories $\rho$ using $\pi^L$,
5:     // and store the trajectories in $\mathcal{D}$ and $\mathcal{R}^L$ respectively.
6:     // After every m timesteps, relabel $\mathcal{D}$ using preference feedback $y$.
7:     // Lower level value function update
8:     **for** each gradient step in t=0 to k **do**
9:         Optimize lower level value function $V^L$ to get $V_m^L$.
10:     // Higher level policy update using DIPPER
11:     **for** each gradient step **do**
12:         // Sample higher level behavior trajectories.
13:         $(\tau^1, \tau^2, y) \sim \mathcal{D}$
14:         Optimize higher level policy $\pi^H$ using equation 9.
15:     // Lower primitive policy update using `RL`
16:     **for** each gradient step **do**
17:         Sample $\rho$ from $\mathcal{R}^L$.
18:         Optimize lower policy $\pi^L$ using SAC.

---

## A.3 RELATED WORK

**Hierarchical Reinforcement Learning.** `HRL` is an elegant framework that promises the intuitive benefits of temporal abstraction and improved exploration (Nachum et al., 2019). Prior research work has focused on developing efficient methods that leverage hierarchical learning to efficiently solve complex tasks (Sutton et al., 1999; Barto & Mahadevan, 2003; Parr & Russell, 1998; Dietterich, 1999). Goal-conditioned `HRL` is an important framework in which a higher-level policy assigns subgoals to a lower-level policy (Dayan & Hinton, 1992; Vezhnevets et al., 2017), which executes primitive actions on the environment. Despite its advantages, `HRL` faces challenges owing to its hierarchical structure, as goal-conditioned `RL` based approaches suffer from non-stationarity in off-policy settings where multiple levels are trained concurrently (Nachum et al., 2018; Levy et al., 2019). These issues arise because the lower level policy behavior is sub-optimal and unstable. Prior works deal with these issues by either simulating optimal lower primitive behavior (Levy et al., 2019), relabeling replay buffer transitions (Nachum et al., 2018; Singh & Namboodiri, 2023b;a), or assuming access to privileged information like expert demonstrations or preferences (Singh et al., 2024; Singh & Namboodiri, 2023a;b). Recent efforts explore intrinsic options to mitigate non-stationarity and infeasible subgoals through self-supervised discovery, such as HIDIO (Zhang et al., 2021), which learns task-agnostic options via entropy minimization on sub-trajectories for diverse exploration in sparse-reward robotics, achieving improved sample efficiency over flat RL. Similarly, HIPPO (Li et al., 2019) unifies low and high-level optimization by enabling joint training to reduce inter-level dependencies, yet it remains reward-dependent and prone to non-stationarity in off-policy updates. In contrast, we propose a novel bi-level formulation to mitigate non-stationarity and regularize the higher-level policy to generate feasible subgoals for the lower-level policy.

Related efforts in automatic curriculum learning employ bi-level optimization to generate adaptive goals or task sequences for lower-level policies, such as (Narvekar & Stone, 2018) who learn curriculum policies in a bi-level MDP to sequence tasks for efficient transfer to a target domain, and (Lewandowski et al., 2022) who use bi-level optimization for multi-objective goal generation in continuous navigation environments to balance exploration, rewards, and non-stationarity through curriculum-driven subgoal refinement. These methods improve sample efficiency in HRL but often rely on heuristics, predefined task graphs, or additional outer-loop RL solvers for curriculum design.

**Behavior Priors.** Some prior work relies on hand-crafted actions or behavior priors to accelerate learning (Nasiriany et al., 2021; Dalal et al., 2021). While these methods can simplify hierarchical learning, their performance heavily depends on the quality of the priors; sub-optimal priors may lead to sub-optimal performance. In contrast, ours is an end-to-end approach that does not require prior specification, thereby avoiding significant expert human effort.

**Preference-based Learning.** A variety of methods have been developed in this area to apply reinforcement learning (RL) to human preference data (Knox & Stone, 2009; Pilarski et al., 2011; Wilson et al., 2012; Daniel et al., 2015), that typically involve collecting preference data from human annotators, which is then used to guide downstream learning. Prior works in this area (Christiano et al., 2017; Lee et al., 2021) first train a reward model based on the preference data, and subsequently employ RL to derive an optimal policy for that reward model. More recent approaches have focused on improving sample efficiency using off-policy policy gradient methods (Haarnoja et al., 2018) to learn the policy. Recently, direct preference optimization based approaches have emerged (Rafailov et al., 2023; 2024; Hejna et al., 2023), which bypass the need to learn a reward model and subsequent RL step, by directly optimizing the policy with a KL-regularized maximum likelihood objective corresponding to a pre-trained model. In this work, we build on the foundational knowledge in maximum entropy RL (Ziebart, 2010), and derive a token-level direct preference optimization (Rafailov et al., 2023; 2024) objective regularized by lower -level primitive, resulting in an efficient hierarchical framework capable of solving complex robotic tasks.

### A.4 PREFERENCE DATA USAGE

Here, we quantify the amount of pair-wise comparisons.

**1. Preference Data Volume:**

In our experiments, we collected approximately 10,000 pairwise trajectory comparisons per environment, depending on task complexity and horizon length. We ensured that these pairwise comparisons represented a diverse array of trajectories, including both near-optimal and suboptimal behaviors, to provide meaningful supervision for the high-level policy. We present the amount of pairwise comparisons (Pairs per Million Env Steps) as follows:

| Environment | Pairwise Comparisons | Env Steps (M) | Pairs per M Steps |
|---|---|---|---|
| Maze | 10,000 | 1.35 | 7,407 |
| Pick & Place | 10,000 | 0.9 | 11,111 |
| Push | 10,000 | 0.775 | 12,903 |
| Kitchen | 10,000 | 0.45 | 22,222 |

Table 1: Pairwise comparisons and efficiency across environments.

### A.5 ADDITIONAL ABLATION EXPERIMENTS

Here, we provide additional ablations to analyze the effect of varying regularization weight $\lambda$ hyper-parameter and $\beta$ hyper-parameter.

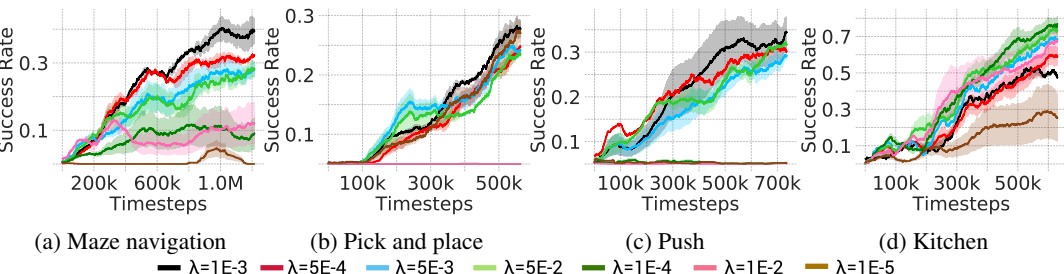

(a) Maze navigation    (b) Pick and place    (c) Push    (d) Kitchen

λ=1E-3    λ=5E-4    λ=5E-3    λ=5E-2    λ=1E-4    λ=1E-2    λ=1E-5

Figure 4: **Regularization weight ablation.** This figure depicts the success rate performance for varying values of the primitive regularization weight $\lambda$. When $\lambda$ is too small, we loose the benefits of primitive-informed regularization resulting in poor performance, whereas too large $\lambda$ values can lead to degenerate solutions. Hence, selecting appropriate $\lambda$ value is essential for accurate subgoal prediction and enhancing overall performance.

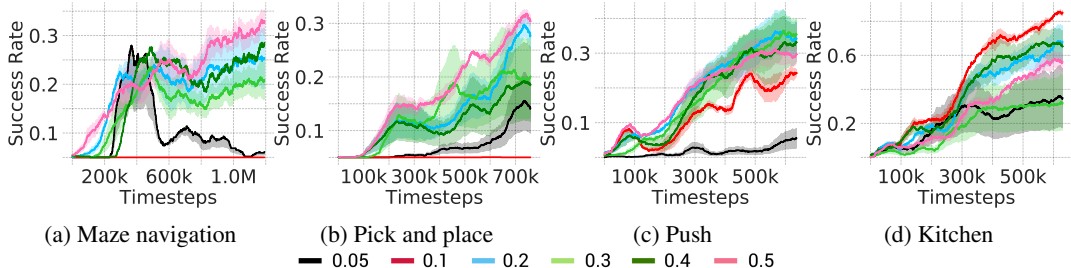

Figure 5: **Max-ent parameter ablation.** This figure illustrates the success rate performance for different values of the max-ent parameter $\beta$ hyper-parameter. This parameter controls the exploration in maximum-entropy formulation. If $\beta$ is too large, the higher-level policy may perform extensive exploration but stay away from optimal subgoal prediction, whereas if $\beta$ is too small, the higher-level might not explore and predict sub-optimal subgoals. Hence, selecting an appropriate $\beta$ value is essential for enhancing overall performance.

## A.6 IMPLEMENTATION DETAILS

We conducted experiments on two systems, each equipped with Intel Core i7 processors, 48GB of RAM, and Nvidia Geforce GTX 1080 GPUs. The experiments included the corresponding timesteps taken for each run. For the environments $(i) - (iv)$, the maximum task horizon $\mathcal{T}$ is set to 225, 50, 50, and 225 timesteps, respectively, with the lower-level primitive allowed to execute for 15, 7, 7, and 15 timesteps. We used off-policy Soft Actor Critic (SAC)(Haarnoja et al., 2018) to optimize the RL objective, leveraging the Adam optimizer(Kingma & Ba, 2014). Both the actor and critic networks consist of three fully connected layers with 512 neurons per layer. The total timesteps for experiments in environments $(i) - (iv)$ are 1.35e6, 9e5, 1.3E6, and 6.3e5, respectively.

For the maze navigation task, a 7-degree-of-freedom (7-DoF) robotic arm navigates a four-room maze with its gripper fixed at table height and closed, maneuvering to reach a goal position. In the pick-and-place task, the 7-DoF robotic arm gripper locates, picks up, and delivers a square block to the target location. In the push task, the arm's gripper must push the square block toward the goal. For the kitchen task, a 9-DoF Franka robot is tasked with opening a microwave door as part of a predefined complex sequence to reach the final goal. We compare our approach with the Discriminator Actor-Critic (Kostrikov et al., 2018), which uses a single expert demonstration. Although this study doesn't explore it, combining preference-based learning with demonstrations presents an exciting direction for future research (Cao et al., 2020).

To ensure fair comparisons, we maintain uniformity across all baselines by keeping parameters such as neural network layer width, number of layers, choice of optimizer, SAC implementation settings, and others consistent wherever applicable. In RAPS, the lower-level behaviors are structured as follows: For maze navigation, we design a single primitive, *reach*, where the lower-level primitive moves directly toward the subgoal predicted by the higher level. For the pick-and-place and push tasks, we develop three primitives: *gripper-reach*, where the gripper moves to a designated position $(x_i, y_i, z_i)$; *gripper-open*, which opens the gripper; and *gripper-close*, which closes the gripper. In the kitchen task, we use the action primitives implemented in RAPS (Dalal et al., 2021).

### A.6.1 ADDITIONAL HYPER-PARAMETERS

Here, we enlist the additional hyper-parameters used in DIPPER:

Table 2: Hyperparameter Configuration

| Parameter | Value | Description |
|---|---|---|
| activation | tanh | activation for reward model |
| layers | 3 | number of layers in the critic/actor networks |
| hidden | 512 | number of neurons in each hidden layer |
| Q_lr | 0.001 | critic learning rate |
| pi_lr | 0.001 | actor learning rate |
| buffer_size | 1E7 | for experience replay |
| clip_obs | 200 | clip observation |
| n_cycles | 1 | per epoch |
| n_batches | 10 | training batches per cycle |
| batch_size | 1024 | batch size hyper-parameter |
| reward_batch_size | 50 | reward batch size for DPO-FLAT |
| random_eps | 0.2 | percentage of time a random action is taken |
| alpha | 0.05 | weightage parameter for SAC |
| noise_eps | 0.05 | std of gaussian noise added to not-completely-random actions |
| norm_eps | 0.01 | epsilon used for observation normalization |
| norm_clip | 5 | normalized observations are cropped to this value |
| adam_beta1 | 0.9 | beta 1 for Adam optimizer |
| adam_beta2 | 0.999 | beta 2 for Adam optimizer |

## A.7 Environment details

### A.7.1 Maze navigation task

In this environment, a 7-DOF robotic arm gripper must navigate through randomly generated four-room mazes. The gripper remains closed, and both the walls and gates are randomly placed. The table is divided into a rectangular $W \times H$ grid, with vertical and horizontal wall positions $W_P$ and $H_P$ selected randomly from the ranges $(1, W - 2)$ and $(1, H - 2)$, respectively. In this four-room setup, gate positions are also randomly chosen from $(1, W_P - 1)$, $(W_P + 1, W - 2)$, $(1, H_P - 1)$, and $(H_P + 1, H - 2)$. The gripper's height remains fixed at table height, and it must move through the maze to reach the goal, marked by a red sphere.

For both higher and lower-level policies, unless Stated otherwise, the environment consists of continuous State and action spaces. The State is encoded as a vector $[p, \mathcal{M}]$, where $p$ represents the gripper's current position, and $\mathcal{M}$ is the sparse maze representation. The input to the higher-level policy is a concatenated vector $[p, \mathcal{M}, g]$, with $g$ representing the goal position, while the lower-level policy input is $[p, \mathcal{M}, s_g]$, where $s_g$ is the subgoal provided by the higher-level policy. The current position of the gripper is treated as the current achieved goal. The sparse maze array $\mathcal{M}$ is a 2D one-hot vector, where walls are denoted by a value of 1 and open spaces by 0.

In our experiments, the sizes of $p$ and $\mathcal{M}$ are set to 3 and 110, respectively. The higher-level policy predicts the subgoal $s_g$, so its action space aligns with the goal space of the lower-level primitive. The lower-level primitive's action, $a$, executed in the environment, is a 4-dimensional vector, where each dimension $a_i \in [0, 1]$. The first three dimensions adjust the gripper's position, while the fourth controls the gripper itself: 0 indicates fully closed, 0.5 means half-closed, and 1 means fully open.

### A.7.2 Pick and place and Push Environments

In the pick-and-place environment, a 7-DOF robotic arm gripper is tasked with picking up a square block and placing it at a designated goal position slightly above the table surface. This complex task involves navigating the gripper to the block, closing it to grasp the block, and then transporting the block to the target goal. In the push environment, the gripper must push a square block towards the goal position. The State is represented by the vector $[p, o, q, e]$, where $p$ is the gripper's current position, $o$ is the block's position on the table, $q$ is the relative position of the block to the gripper, and $e$ contains the linear and angular velocities of both the gripper and the block.

The higher-level policy input is the concatenated vector $[p, o, q, e, g]$, where $g$ denotes the target goal position, while the lower-level policy input is $[p, o, q, e, s_g]$, with $s_g$ being the subgoal provided by

the higher-level policy. The current position of the block is treated as the achieved goal. In our experiments, the dimensions for $p$, $o$, $q$, and $e$ are set to 3, 3, 3, and 11, respectively. The higher-level policy predicts the subgoal $s_g$, so the action and goal space dimensions align. The lower-level action $a$ is a 4-dimensional vector, where each dimension $a_i$ falls within the range $[0, 1]$. The first three dimensions adjust the gripper's position, and the fourth controls the gripper itself (0 for closed, 1 for open). During training, the block and goal positions are randomly generated, with the block always starting on the table and the goal placed above the table at a fixed height.

## A.8    ENVIRONMENT VISUALIZATIONS

Here, we provide some visualizations of the agent successfully performing the task.

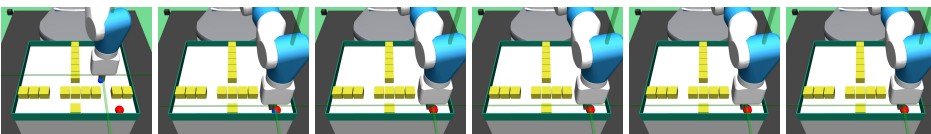

Figure 6: **Maze navigation task visualization**: The visualization is a successful attempt at performing maze navigation task

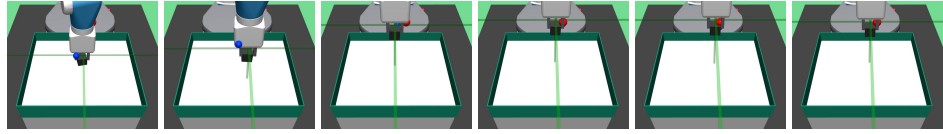

Figure 7: **Pick and place task visualization**: This figure provides visualization of a successful attempt at performing pick and place task

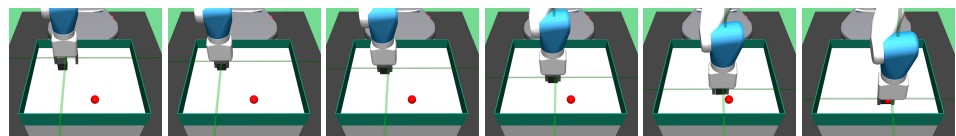

Figure 8: **Push task visualization**: The visualization is a successful attempt at performing push task

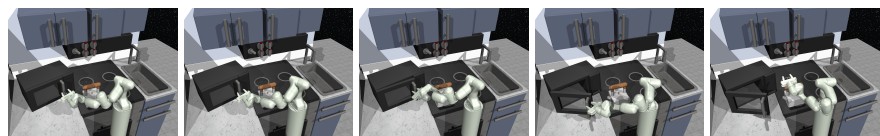

Figure 9: **Kitchen task visualization**: The visualization is a successful attempt at performing kitchen task

