# OpenReview forum: "Direct Preference Optimization for Primitive-Enabled Hierarchical RL: A Bilevel Approach"
_ICLR.cc/2026/Conference — ICLR 2026 Poster_

### Official Review · Reviewer_fdjG · 2025-10-19

**Soundness:** 2
**Presentation:** 3
**Contribution:** 3
**Rating:** 4
**Confidence:** 4

**Summary:**

The authors propose DIPPER, a hierarchical reinforcement learning (HRL) framework that formulates HRL as a bi-level optimization problem. In this formulation, the higher-level “teacher” policy and lower-level “student” policy are jointly optimized while explicitly modeling their interdependence. To mitigate the non-stationarity caused by the evolving lower-level policy, the authors leverage Direct Preference Optimization (DPO) to train the higher-level policy on stationary human preference data rather than environment rewards. Additionally, they introduce a lower-level value function regularization term that encourages the higher-level policy to propose feasible and achievable subgoals.

**Strengths:**

The paper provides a rigorous bi-level mathematical formulation of hierarchical reinforcement learning (HRL).

The use of real human preference data is commendable. Conducting human-in-the-loop experiments introduces significant complexity and time overhead, yet provides stronger evidence for the method’s practical relevance

The authors empirically evaluate whether DIPPER actually addresses the two challenges they identify, non-stationarity and infeasibility of subgoal, rather than assuming these issues are solved. This careful, hypothesis-driven evaluation is a strong aspect of the paper.

They explore how varying key hyperparameters (λ, β) affects performance, showing an understanding of what aspects drive DIPPER’s behavior.

**Weaknesses:**

**Stationarity of Human Preferences**

The paper assumes that human preferences mitigate non-stationarity, yet human preferences themselves are not necessarily stationary.
Prior work suggests that people’s judgments can change over time [1], particularly when confidence is low.

Question: How would DIPPER handle cases where human preferences shift if the same trajectories are presented multiple times? Would the algorithm adapt?

**Missing Related Work**

The related work section omits relevant approaches in automatic curriculum learning that use bi-level optimization to generate goals for lower-level policies.
Examples include Narvekar and Stone (2019) [2] and Muslimani et al. (2023) [3].
These should be discussed to contextualize DIPPER within existing curriculum learning frameworks.

**Failure Cases and Environment Suitability**

DIPPER fails in the simple maze task. The paper does not explain why.

Question: What types of environments does DIPPER work best in?

In environments where DIPPER outperforms baselines (Pick & Place and Push), success rates remain low (around 30%).

Question: Why does performance plateau at this relatively low level? Would longer training or additional preference data improve results?


**Baselines**

The authors remove the HER component from PIPER “for fairness,” but this substantially alters the original algorithm.
This change may make the comparison less meaningful.

Question: How would DIPPER perform against the original PIPER implementation (with HER) as described in the source paper?

The authors mention tuning DIPPER’s hyperparameters using grid search, but it is unclear whether the same effort was made for the baselines.
 This introduces a potential unfair advantage if baselines are not tuned equivalently.

**Evaluation Metrics and Interpretation**

The “lower Q-function metric” measures Q-values for subgoals proposed by the higher-level policy. However, this could be inflated if the higher policy proposes easy subgoals.

Question: Do the proposed metrics actually capture whether DIPPER learns progressively more difficult subgoals over time?


**Human Preference Data**

The paper provides limited details about the human annotators.

Questions:

Who were the annotators (experts or laypeople)?

Were they trained or provided with guidance?

What were their demographic backgrounds?

**Most importantly, was this data collection ethics-approved?**

These details are crucial to assess the reliability and reproducibility of the preference data.


**Sources:**
[1]Visser et al (2005) https://pubmed.ncbi.nlm.nih.gov/16022057/

[2] Narvekar and Stone (2019) https://arxiv.org/pdf/1812.00285

[3] Muslimani et al (2023) https://arxiv.org/pdf/2204.11897

**Questions:**

I included my questions in the weaknesses section.

**Details Of Ethics Concerns:**

The authors use human subjects, but I cannot determine whether the data collected was approved by an ethics committee.
If the study was ethics approved (which the authors can address during the rebuttal), then I do not believe any further review in this issue is necessary.

---

> ### Author Response · Authors · 2025-11-21
> **Response to Reviewer fdjG [Part 1]**
>
> We thank the reviewer for their thoughtful and constructive feedback, which has helped us improve the clarity and positioning of our work. Below we address each point in turn.
>
> > **Strength:** The use of real human preference data is commendable. Conducting human-in-the-loop experiments introduces significant complexity and time overhead, yet provides stronger evidence for the method’s practical relevance
>
> **Response:** We apologise for the confusion and take this opportunity to clarify and udpate our paper.
>
> **Correction regarding Human Preference Data:** We appreciate the reviewer's recognition of the complexity involved in human-in-the-loop experiments. For clarity and correctness, we would like to refine the terminology used in the current draft. Although the text occasionally referred to “human annotations,” our experiments in fact use **synthetic preferences** generated from ground-truth environment rewards using a Bradley–Terry model [1]. Given two trajectories A and B with cumulative rewards $r_A$ and $r_B$,  the preference probability is P(prefer A over B) = $\frac{1}{1 + \exp(r_B - r_A)}$. Thus, in the current version of the work, human feedback itself is not the main contribution. Instead, the focus is on preference-based hierarchical RL using a clean, controlled preference signal. This synthetic setup is standard in preference-based RL [1,2,3], and it allows us to directly validate our bi-level formulation and value regularization without confounding factors from annotator variability.
>
> - The core contributions of DIPPER: (i) mitigating non-stationarity in hierarchical RL and (ii) reducing infeasible subgoal generation, remain unchanged under synthetic preferences, since the preference dataset is stationary and derived from the underlying environment reward function.
>
> - **Revised Version of the Paper:** We have updated Sections 1 and 4 to explicitly describe the synthetic preference-generation procedure and to avoid references suggesting human annotation. The Ethics Statement now clearly notes that no human subjects were involved and no approvals were required, as all preferences are generated from environment rewards. We hope this clarification resolves the concern about human data.
>  Our supplementary implementation code included scripts for synthetic generation of preference dataset, ensuring reproducibility.
>
> [1] Christiano et al. "Deep reinforcement learning from human preferences" arXiv preprint arXiv:1706.03741, 2017.
>
> [2] Lee et al. "B-Pref: Benchmarking Preference-Based Reinforcement Learning" arXiv preprint arXiv:2111.03026, 2021.
>
> [3] Bewley et al. "Interpretable Preference-based Reinforcement Learning with Tree-Structured Reward Functions" arXiv preprint arXiv:2112.11230, 2021.
>
> ---
>
> #### Stationarity of Human Preferences
>
> > **Weakness 1:** The paper assumes that human preferences mitigate non-stationarity, yet human preferences themselves are not necessarily stationary. Prior work suggests that people’s judgments can change over time [1], particularly when confidence is low.
>
> > **Question 1:** How would DIPPER handle cases where human preferences shift if the same trajectories are presented multiple times? Would the algorithm adapt?
>
> **Response:** We thank the reviewer for this insightful point. In DIPPER (as explained earlier), the key assumption is that the preference dataset is derived synthetically from the ground-truth rewards, which remains stationary with respect to the lower-level policy.
>
> We agree that in prctice, preferences may evolve over time (e.g., with learning, fatigue, or changing objectives), which could introduce additional non-stationarity. Extending DIPPER to such settings is a natural and interesting direction. One possible approach would be:
>
> - Developing an online variant of DIPPER that periodically updates or re-weights the preference dataset as new (potentially shifted) feedback is collected.
> - Incorporating recent formulations such as NS-DPO~[4], which explicitly model time-dependent or drifting preferences, into the higher-level objective to better accommodate preference drift.
>
> [4] Son et al. "Right Now, Wrong Then: Non-Stationary Direct Preference Optimization under Preference Drift" arXiv preprint arXiv:2407.18676, 2024.
>
> ---
>
> #### Missing Related Work
>
> > **Weakness 1:** The related work section omits relevant approaches in automatic curriculum learning that use bi-level optimization to generate goals for lower-level policies. Examples include Narvekar and Stone (2019) [2] and Muslimani et al. (2023) [3]. These should be discussed to contextualize DIPPER within existing curriculum learning frameworks.
>
> **Response:** We thank the reviewer for pointing out these relevant works on automatic curriculum learning with bi-level optimization, which indeed provide important context for subgoal generation in HRL. We have added these works with explanation in the detailed related work in the revised manuscript (Appendix A.3).
>
> ---

---

> > ### Author Response · Authors · 2025-11-21
> > **Response to Reviewer fdjG [Part 2]**
> >
> > #### Failure Cases and Environment Suitability
> >
> > > **Weakness 1:** DIPPER fails in the simple maze task. The paper does not explain why.
> >
> > > **Question 1:** What types of environments does DIPPER work best in?
> >
> > **Response:** We would like to clarify that while DIPPER does not fail (it achieves approximately 35% success rate, Figure 2), it is indeed outperformed by certain baselines like RAPS, HAC and SAGA in simple maze task.
> >
> > **Why some Baselines Outperform in Simple Maze:** Baselines like RAPS, HAC, SAGA leverage additional information or assumptions unavailable to DIPPER, which are straightforward to implement in low-complexity navigation but impractical in complex manipulation settings:
> > 1. RAPS assumes access to pre-constructed behavior priors at the lower level (such as hand-crafted policies for maze navigation). These priors require domain-specific engineering that scales poorly to complex manipulation tasks like franka kitchen.
> > 2. HAC simulates optimal lower-level primitives, which is feasible for simpler maze-like environments, but infeasible in high-dimensional manipulation tasks.
> > 3. SAGA assumes a state-conditioned discriminator for subgoal selection, which is easier to train for low-variance, structured tasks like mazes but is harder to train for complex manipulation tasks.
> >
> > In contrast, DIPPER only assumes access to preference dataset and uses learned hierarchies via bi-level DPO and value regularization, without assuming access to behavior priors, discriminators, or optimal behavior simulation.
> >
> > **Response to Question 1:** In the simple maze navigation task (where it is easier to design behavior priors and train near-optimal lower-level behaviors), approaches like RAPS or HAC would work better than DIPPER. However, in complex tasks like long-horizon sparse-reward manipulation like pick-and-place, Franka Kitchen, etc. (where designing near-optimal behavior priors is either infeasible, requires non-trivial human effort, or is computationally expensive), employing DIPPER would be ideal as it makes minimal assumptions about the environment and does not require access to such privileged information. Indeed, as seen in Figure 2, DIPPER is able to outperform these baselines in such harder tasks (pick and place, bin and Franka kitchen).
> >
> > ---
> >
> > > **Weakness 2:** In environments where DIPPER outperforms baselines (Pick & Place and Push), success rates remain low (around 30%).
> >
> > > **Question 2:** Why does performance plateau at this relatively low level? Would longer training or additional preference data improve results?
> >
> >
> > **Response:** In the Pick & Place and Push environments, DIPPER achieves around 30—35% success rates that, while modest, still outperform the baselines. This plateau reflects the inherent difficulties of long-horizon, sparse-reward manipulation tasks, which require extensive exploration and stable hierarchical credit assignment. In DIPPER, the preference-based learning mechanism helps align high-level policies with effective subgoals, but the low signal from sparse rewards limits the granularity of feedback for effective training.
> >
> > **Empirical Analysis:** We empirically observed that adding more preference data does increase the success rate upto a point and then plateaus. Further, although we report the current results upto around ~1M steps for maze and pick and place, and ~600k for push and kitchen tasks due to resource constraints, we found that longer training does improve success rates in the pick-and-place task. Also, as kindly suggested by the reviewer, we are currently imcorporating the orthogonal approch HER to DIPPER to further improve performance, and we will add these new results to the final manuscript.
> >
> > ---
> >
> > #### Baselines
> >
> > > **Weakness 1:** The authors remove the HER component from PIPER “for fairness,” but this substantially alters the original algorithm. This change may make the comparison less meaningful.
> >
> > > **Question 1:** How would DIPPER perform against the original PIPER implementation (with HER) as described in the source paper?
> >
> > **Response:** To ensure a meaningful comparison isolating the effects of preference-based learning and non-stationarity mitigation, we ablated HER from PIPER, as it is an orthogonal exploration technique that can be applied to any sparse-reward method, including DIPPER. For completeness, we are incorporating HER into DIPPER and plan to report the updated results (PIPER, PIPER w\o HER, DIPPER, DIPPER w\ HER) and add them to the final manuscript.
> >
> > ---

---

> > ### Comment · Reviewer_fdjG · 2025-11-21
> > **Score change.**
> >
> > Based on the authors’ clarification, I unfortunately have to lower my score. Originally, one of the primary strengths of this work was my understanding that the results were based on real human feedback. However, it appears that only simulated feedback is used. I strongly believe that evaluating preference-learning algorithms solely with synthetic feedback derived from the ground-truth reward is insufficient. Humans are inherently noisy and inconsistent, and thus meaningful validation requires testing with actual human raters to understand how the method performs under realistic conditions.

---

> ### Author Response · Authors · 2025-11-21
> **Response to Reviewer fdjG [Part 3]**
>
> > **Weakness 2:** The authors mention tuning DIPPER’s hyperparameters using grid search, but it is unclear whether the same effort was made for the baselines. This introduces a potential unfair advantage if baselines are not tuned equivalently.
>
> **Response:** To ensure fair comparisons, we applied the hyper-parameters from original paper implementations wherever available, and used the same grid search procedure for all baselines (e.g., PIPER, HAC, SAGA, RAPS, flat SAC), where we tune key hyperparameters such as learning rates, entropy coefficients etc over multiple seeds. The results are reported using the best configuration, and we assure the reviewer that there was no unfair advantage provided to DIPPER.
>
> ---
>
> #### Evaluation Metrics and Interpretation
>
> > **Weakness 1:** The “lower Q-function metric” measures Q-values for subgoals proposed by the higher-level policy. However, this could be inflated if the higher policy proposes easy subgoals.
>
> **Response:** We propose the *lower Q-function metric* to assess if DIPPER mitigates $(i)$ non-stationarity, and $(ii)$ infeasible subgoal generation issues in HRL. We agree with the reviewer that if the predicted subgoals are easy, the Q-values could be inflated. However, this does not occur in DIPPER as explained below:
>
> Consider our principled bi-level formulation in Eq. 7:
>
> \begin{align}
>     \mathcal{L}\_{\mathcal{O}} & = -\mathbb{E}\_{(\tau^1, \tau^2, y)\sim\mathbb{D}} \biggl[ \log \sigma \biggl(\sum\_{t=0}^{T-1}\bigl(\underbrace{\beta \log \pi\_{\*}^{H}(g\_t^1|s\_t^1,g^\*)
>       - \beta \log \pi\_{\*}^{H}(g\_t^2|s\_t^2,g^\*)}\_{\text{PbL term}} \nonumber
>      - \lambda ( \underbrace{(V^L(s\_t^1,g\_t^1)  - \\!V^L\_\*(s\_t^1,g\_t^1) - (V^L(s\_t^2,g\_t^2)  - V^L\_\*(s\_t^2,g\_t^2)) )}\_{\text{Regularization term}}\bigr)\biggr)\biggr].
> \end{align}
> Here, the first *PbL term* ensures that the higher level policy learns to achieve the final goal using DPO, while the *regularization term* ensures that the subgoals remain achievable for the lower level. Since this objective ensures progress towards final goal while ensuring feasible subgoal predition, it avoids degenerate solutions, like the higher level policy only predicting easy but feasible subgoals. Thus, in case of DIPPER, *lower Q-function metric* remains a good metric to measure if the proposed approach mitigates non-stationarity and infeasible subgoal generation in HRL.
>
> However, for approaches which e.g. only predict easy subgoals (leading to degenerate solutions without progressing towards the final goal), Q-values can indeed over-inflate. Therefore, in such cases, we require both the success rates (Figure 2) and Q-value metric values (Figure 3) for complete analysis of their efficacy.
>
> ---
>
> > **Question 1:**  Do the proposed metrics actually capture whether DIPPER learns progressively more difficult subgoals over time?
>
>
> **Response:** Yes, our proposed metrics (subgoal distance and lower Q-function, Fig. 3), along with our success rate analysis in Figure 2, captures that DIPPER learns progressively more difficult subgoals over time. As seen from Figure 2, the success rates are initially low, which indicates that the subgoals are close and simple for basic exploration, and are unable to reach the final goal. As the training advances, the success rates increase, which implies that the higher-level policy predicts harder, longer-range subgoals until the final goal is reached. In Fig. 3, the subgoal distance metric (Row 1) and the lower Q-value metric (Row 2) ensure subgoals stay achievable despite being increasingly harder, outperforming the baselines. Together, these show that DIPPER is able to achieve progressively harder subgoals, while the subgoals always remain reachable for the lower-level primitive.
>
> ---
>
> #### Human Preference Data
>
> > **Weakness 1:** The paper provides limited details about the human annotators.
>
> **Response:** As clarified above, our experiments use synthetically generated preference data based on cumulative environment rewards, not human annotators. We have updated the manuscript to:
> 1. Remove ambiguous language suggesting human annotation,
> 2. Describe the synthetic preference-generation procedure in more detail, and
> 3. Clarify in the Ethics Statement that no human subjects were involved.
>
> ---
> We hope these responses clarify the concerns. Please let us know if you need further clarifications.

---

> ### Author Response · Authors · 2025-11-22
> **Author's Response**
>
> We sincerely thank the reviewer for taking the time to review our paper.
>
> In this submission, our goal is to study the **algorithmic and structural aspects of hierarchical preference-based RL** under a controlled setting. For this reason, we follow common practice in the preference-RL literature and used synthetic preferences derived from the ground-truth reward (e.g., Christiano et al., 2017; Lee et al., 2021; Bewley et al., 2021). This allows us to isolate the effects of our bi-level formulation and value regularization without additional variability from human raters.
>
> We respect the reviewer’s emphasis on human-based evaluation. We believe the current synthetic-preference experiments provide meaningful evidence about the behavior of DIPPER and its contributions on the algorithmic side of preference-based hierarchical RL.

---

> > ### Comment · Reviewer_fdjG · 2025-11-23
> >
> > I understand that it is important to do experiments in a controlled setting. However, there should be at least one experiment (even in a simpler environment) that demonstrates the method can work with humans.
> >
> > I'll also add that Christiano et al., 2017 did a large set of human experiments and Lee et al., 2021 at least had experiments where the authors provided feedback. Moreover, recent PbRL works are incorporating human experiments. See the papers below for some examples.
> >
> > [1] Hejna, J., & Sadigh, D. (2022). Few-Shot Preference Learning for Human-in-the-Loop RL. Proceedings of the 6th Annual Conference on Robot Learning.
> >
> > [2] White, D., Wu, M., Novoseller, E., Lawhern, V. J., Waytowich, N., & Cao, Y. (2024). Rating-Based Reinforcement Learning. Proceedings of the AAAI Conference on Artificial Intelligence.
> >
> > [3] Muslimani, C., & Taylor, M. E. (2025). Leveraging sub-optimal data for human-in-the-loop reinforcement learning. International Conference on Learning Representations.

---

> > > ### Author Response · Authors · 2025-11-23
> > >
> > > We sincerely appreciate the reviewer’s perspective and thank you for the references. Indeed, evaluating with real human raters is an important next step.

---

### Official Review · Reviewer_DfyA · 2025-10-26

**Soundness:** 3
**Presentation:** 3
**Contribution:** 3
**Rating:** 8
**Confidence:** 4

**Summary:**

The authors present DIPPER, a hierarchical RL method that directly optimizes environment reward and human preferences. Given human preferences, the authors derive a bi-level optimization objective that train the policy to increase the likelihood of the preferred trajectories with the high-level policy while simultaneously updating the low-level policy to try to maintain value improvement for the same objective. The specific objective is derived from DPO, and helps mitigate non-stationarity common to most HRL methods.

**Strengths:**

**Comparisons:** The authors compare against a wide array of relevant baselines, demonstrating great performance against said baselines on manipulation tasks especially.

**Motivation:** Non-stationarity in HRL is a big issue and this paper presents a well-motivated solution for it.

**Experiments:** The experiments are performed on tasks well-suited for HRL, and the analysis on goal distance prediction against HIER and HAC demonstrates that DIPPER’s objective encourages sampling reachable goals for the lower-level policy.

**Clarity:** The paper is overall quite clear and the walkthrough of how to obtain the objective was both interesting and easy to read.

**Weaknesses:**

**Figures**: All of the results figures have small font that make them harder to read, and are also clearly image files put into overleaf instead of vectorized PDFs. The results figures should have thicker lines for each baseline, more spacing between baselines on the legend (in fig 2), and larger font for the x and y axes labels and ticks labels.

**Annotation cost:** As authors admit, there is a high annotation cost to obtaining labels with human prediction.

**Experiments:** Given the fact that the authors have human annotations, it seems that more challenging tasks could’ve been demonstrated in the experiments. This is not a reason to reject the paper, however I will list this as a slight weakness of the paper.

**Minor issues:**

- There’s related work which also mitigates non-stationary and infeasible subgoal generation by modeling *intrinsic* options: e.g., https://arxiv.org/abs/2101.06521, some comparison against this work and any follow-ups in the related works section would be beneficial
- There’s also related work on unifying low level and high level policy optimization: https://sites.google.com/view/hippo-rl
- L363: “we maintain empirical consistency across all baselines to ensure fair comparisons” — what does this actually mean? be specific here

**Questions:**

Instead of a subgoal distance metric (fig 3), why not directly measure the success rate of the lower level policy’s ability to reach goals? This is the actual metric that matters for addressing the subgoal feasability problem, right?

Re: annotation cost, prior work in reward learning has demonstrated that VLMs can be useful for obtaining preferences in place of humans: https://rlvlmf2024.github.io/, have the authors looked into this or tried out similar approaches at a small scale?

---

> ### Author Response · Authors · 2025-11-21
> **Response to Reviewer DfyA [Part 1]**
>
> We would like to express our gratitude to the reviewer for dedicating their valuable time and effort towards evaluating our manuscript, which has allowed us to strengthen the manuscript. We deeply appreciate the insightful feedback provided, and we have thoroughly responded to reviewer's inquiries in the responses provided below.
>
> ## Weaknesses
>
> > **Figures:** All of the results figures have small font that make them harder to read, and are also clearly image files put into overleaf instead of vectorized PDFs. The results figures should have thicker lines for each baseline, more spacing between baselines on the legend (in fig 2), and larger font for the x and y axes labels and ticks labels.
>
> **Response:** We thank the reviewer for the suggestions, and have fixed the previous issues (line size, spacing, font size of ticks and labels, etc) in the figures in the revised manuscript.
>
> ---
>
> > **Annotation cost:** As authors admit, there is a high annotation cost to obtaining labels with human prediction.
>
> > **Experiments:** Given the fact that the authors have human annotations, it seems that more challenging tasks could’ve been demonstrated in the experiments. This is not a reason to reject the paper, however I will list this as a slight weakness of the paper.
>
> **Response:** Thank you for your comments and we apologise for the oversight regarding annotation cost.
>
> **Clarification on annotation cost.** We wish to rectify some imprecise phrasing in the draft realted to human annotations, our experiments use synthetic preference data generated from ground-truth environment rewards. Concretely, we simulate human judgments with a Bradley–Terry model [1], assigning preferences based on cumulative rewards for two trajectories A and B:
> $P(\text{prefer A over B}) = \frac{1}{1 + \exp(r_B - r_A)}$,
> where $r_A$ and $r_B$ are the total rewards. This proxy is standard in preference-based RL [1,2,3] and allows us to study the bi-level formulation and value regularization in a controlled setting without incurring human annotation cost. We have revised the manuscript (Sections 1 and 4) and the Ethics Statement to explicitly state that no human subjects were involved and that preferences are synthetically generated from environment rewards.
>
> **On task complexity and experimental scope.** We agree that, if real human annotations were available at scale, it would be compelling to demonstrate DIPPER on even more complex tasks. In this work, we only focus on widely used long-horizon, sparse-reward benchmarks (maze navigation, pick-and-place, push, Franka Kitchen), which already expose the **key challenges of hierarchical non-stationarity and infeasible subgoal generation** in a reproducible setting. Our synthetic-preference experiments is a first step that isolates the algorithmic behavior of DIPPER under well-controlled preferences.
>
> [1] Christiano et al. "Deep reinforcement learning from human preferences" arXiv preprint arXiv:1706.03741, 2017.
>
> [2] Lee et al. "B-Pref: Benchmarking Preference-Based Reinforcement Learning" arXiv preprint arXiv:2111.03026, 2021.
>
> [3] Bewley et al. "Interpretable Preference-based Reinforcement Learning with Tree-Structured Reward Functions" arXiv preprint arXiv:2112.11230, 2021.
>
> ---
>
>
> > **Minor issues 1:** There’s related work which also mitigates non-stationary and infeasible subgoal generation by modeling intrinsic options: e.g., https://arxiv.org/abs/2101.06521, some comparison against this work and any follow-ups in the related works section would be beneficial
>
> > **Minor issues 2:** There’s also related work on unifying low level and high level policy optimization: https://sites.google.com/view/hippo-rl
>
> **Response:** We thank the reviewer for pointing out these references, which enrich the discussion on intrinsic options and unified hierarchical optimization in HRL. We have incorporated comparisons to these references in the revised related work section.
>
> ---
>
> > **Minor issues 3:** L363: “we maintain empirical consistency across all baselines to ensure fair comparisons” — what does this actually mean? be specific here
>
> **Response:** We apologize for the ambiguity in this statement and clarify here. To ensure fair comparisons, we kept identical environments and training conditions (maze, pick and place, push and kitchen), copied the hyper-parameters from original paper implementations wherever available, and otherwise used grid search procedure to get hyper-parameters for all baselines (e.g., PIPER, HAC, SAGA, RAPS, and flat SAC). The key hyperparameters such as learning rates, entropy coefficients etc are tuned over multiple seeds, and the results are reported using the best configuration of the hyper-parameters. We will add this explanation in the final manuscript.
>
> ---

---

> ### Author Response · Authors · 2025-11-21
> **Response to Reviewer DfyA [Part 2]**
>
> > **Question 1:** Instead of a subgoal distance metric (fig 3), why not directly measure the success rate of the lower level policy’s ability to reach goals? This is the actual metric that matters for addressing the subgoal feasability problem, right?
>
> **Response:**  We appreciate the insightful question. The success rate of the lower-level policy measures the agent's ability to reach the final goal, as shown in Figure 2. However, *subgoal feasibility* specifically refers to the higher-level policy's ability to propose subgoals that are achievable by the lower-level policy. It is possible for the success rate to be relatively high while the higher-level policy predicts infeasible subgoals (for example, the agent might complete the task by bypassing infeasible intermediate subgoals and reach the final goal, especially if the task is too easy). Therefore, instead of the success rate, we use the subgoal distance and the lower Q function metrics to measure the subgoal feasibility in Figure 3. Together with the success rate performance in Figure 2, these metrics accurately analyze the approach's progress towards the final goal, while ensuring achievable subgoals prediction.
>
> ---
>
> > **Question 2:** Re: annotation cost, prior work in reward learning has demonstrated that VLMs can be useful for obtaining preferences in place of humans: https://rlvlmf2024.github.io/, have the authors looked into this or tried out similar approaches at a small scale?
>
> **Response:** As clarified  earlier response and the revised manuscript, DIPPER's experiments rely on synthetic preferences derived from ground-truth environment rewards via the Bradley-Terry model to efficiently generate a large-scale dataset simulating realistic human judgments, without human or VLM annotation costs.
>
> Although we have not yet explored VLMs for generating preferences from text goals and visual observations, we thank the reviewer for this suggestion. We agree that this represents a promising direction for future extensions of DIPPER, potentially integrating VLM feedback to handle diverse, long-horizon manipulation beyond our current benchmarks.
>
> ---
>
> We hope these responses clarify the concerns. Please let us know if you need further clarifications.

---

### Official Review · Reviewer_ywxX · 2025-10-31

**Soundness:** 2
**Presentation:** 2
**Contribution:** 2
**Rating:** 4
**Confidence:** 2

**Summary:**

This paper has proposed a novel hierarchical reinforcement learning framework, which employs preference-based learning in the high-level policy to mitigate the non-stationary issue in hierarchical policy learning. The proposed framework is evaluated in a set of simulated long-horizon tasks.

**Strengths:**

1.	The research problem of non-stationarity is significantly important in the hierarchical reinforcement learning domain.

**Weaknesses:**

1.	The technical contribution of this work is low, which is a direct application of preference-based learning to high-level policy optimization.

2.	Experiments are limited to simulated environments. Demonstrations in real robots or transfer scenarios would significantly strengthen the empirical validation.

**Questions:**

1.	How sensitive is DIPPER to the scale or bias of human preferences? Would synthetic or learned preferences generalize effectively?

2.	Could DIPPER be extended to fully autonomous preference learning (e.g., self-generated ranking signals) without human input?

---

> ### Author Response · Authors · 2025-11-21
> **Response to Reviewer ywxX [Part 1]**
>
> We would like to express our gratitude to the reviewer for dedicating their valuable time and effort towards evaluating our manuscript, which has allowed us to strengthen the manuscript. We deeply appreciate the insightful feedback provided, and we have thoroughly responded to reviewer's inquiries in the responses provided below.
>
> ## Weaknesses
>
> > **Weakness 1:** The technical contribution of this work is low, which is a direct application of preference-based learning to high-level policy optimization.
>
> **Response:** We clarify DIPPER's contributions and highlight DIPPER's novelty below.
>
> **DIPPER's Contributions:** We would like to clarify that DIPPER introduces three key technical contributions tailored to HRL challenges:
>
> 1. **Novel Bi-level Formulation:** We propose a novel bi-level formulation for hierarchical RL that explicitly models interdependencies between hierarchical policies (Eq. 8), deriving a lower level value function based regularization to regularize the higher level policy to generate feasible subgoals, a gap not addressed in prior preference based learning approaches in HRL like PIPER [1].
> 2. **Mitigating HRL Non-Stationarity**: We decouple higher-level learning from non-stationary lower-level rewards via stationary preferences with direct preference optimization for efficiency and stability (Figure 2).
> 3. **Novel Metrics:** We introduce novel metrics for evaluating non-stationarity and subgoal feasibility in HRL (subgoal distance and lower Q-value metric), which was earlier absent in prior works (Figure 3).
>
> These key contributions are combined to solve longstanding HRL issues (non-stationarity, infeasible subgoal generation) in a principled manner. We hope that these points, along with our empirical analysis in sparse reward environments clarifies DIPPER's novelty and  contributions.
>
> [1] Singh et al. "PIPER: Primitive-Informed Preference-based Hierarchical Reinforcement Learning via Hindsight Relabeling" arXiv preprint arXiv:2404.13423, 2024.
>
> ---
>
> > **Weakness 2:** Experiments are limited to simulated environments. Demonstrations in real robots or transfer scenarios would significantly strengthen the empirical validation.
>
> **Response:** In this work, we have conducted simulated experiments on four Mujoco benchmarks: maze navigation, pick and place, push and kitchen environment commonly used in standard HRL literature, to validate DIPPER's core contributions: mitigating HRL non-stationarity and infeasible subgoals generation via DPO based bi-level optimization framework for solving long-horizon, sparse-reward tasks, yielding 35-40% success improvements over baselines (Fig. 2).
>
> We agree with the reviewer that demonstrations in real robots or transfer scenarios would significantly strengthen the results. However, we focussed primarily on the empirical validation of our DPO based bi-level formulation in simulated environments in this paper, and we plan to conduct sim-to-real transfer and real robot experiments in future work.
>
> ---

---

> ### Author Response · Authors · 2025-11-21
> **Response to Reviewer ywxX [Part 2]**
>
> > **Question 1:** How sensitive is DIPPER to the scale or bias of human preferences? Would synthetic or learned preferences generalize effectively?
>
> **Response:** In our implementation, we train DIPPER's higher-level policy using direct preference optimization (DPO) on synthetic preferences generated from ground-truth environment rewards via the Bradley-Terry model, where pairwise preferences are derived from cumulative rewards: P(prefer A over B) = $\frac{1}{1 + \exp(r_B - r_A)}$, where $r_A$ and $r_B$ are total rewards for trajectories A and B. This setup renders DIPPER insensitive to scale or bias in human preferences, as no human annotations are involved; instead, preferences are noise-free and stationary, derived directly from fixed environment rewards independent of policy evolution or human variability.
>
> **Generalization of Synthetic or Learned Preferences:** Synthetic preferences in DIPPER generalize effectively across diverse sparse-reward robotic tasks (e.g., maze navigation, pick-and-place, push, Franka Kitchen), achieving up to 40% success rate improvements over baselines. Further, our ablations confirm robustness: we show in our noise sensitivity analysis (Table 2) that DIPPER remains robust under noisy preference feedback and is able to maintain robust performance across environments.
>
> ---
>
> > **Question 2:** Could DIPPER be extended to fully autonomous preference learning (e.g., self-generated ranking signals) without human input?
>
> **Response:** Yes, DIPPER can be readily extended to autonomous preference learning by replacing human preferences with self-generated ranking signals. In fact, to efficiently generate a large-scale preference dataset while simulating realistic human judgments, we used synthetic preferences derived from the ground-truth environment rewards via the Bradley-Terry model. This approach is a standard proxy in preference-based RL literature [1], allowing us to validate DIPPER's bi-level formulation via fully autonomous preference learning without the high cost of human annotation. The results in Figure 2 shows that under this setting, DIPPER achieves up to 40% success rate improvements over baselines.
>
> [1] Christiano et al. "Deep reinforcement learning from human preferences" arXiv preprint arXiv:1706.03741, 2017.
>
> ---
>
> We hope these responses clarify the concerns. Please let us know if you need further clarifications.

---

### Official Review · Reviewer_W15h · 2025-11-11

**Soundness:** 3
**Presentation:** 3
**Contribution:** 3
**Rating:** 8
**Confidence:** 4

**Summary:**

This introduces a hierarchical RL framework (DIPPER) aimed to tackles two HRL issues: 1. non-stationarity of the higher-level environment and 2. infeasible subgoal generation. The framework is composed of a high level goal-conditioned HRL with Direct Preference Optimization (DPO) and standard RL at the low level. This work formalize HRL as a constrained bi-level optimization problem.

Experiments on four challenging sparse-reward navigation and manipulation tasks (random mazes, pick-and-place, push, and Franka kitchen) show that DIPPER substantially outperforms both standar and hierarchical baselines, including prior preference-based HRL methods. This work also uses subgoal distance and low-level Q value to empirically support claims about reduced non-stationarity and improved subgoal feasibility.

**Strengths:**

1. Originality: The paper gives a clean bi-level optimization formulation of goal-conditioned HRL, rewriting the hierarchy as an upper-level problem with a constraint that the lower-level policy is (locally) optimal. In contrast to PIPER, which learns a reward model and then uses RL/RLHF on top of it, this work directly trains the high-level policy via DPO on a preference dataset, which allows the framework to avoid modeling explicit rewards and avoids having a second RL loop.

2. Significance: This paper provides a preference-based, DPO-driven recipe with a clear regularizer derived from a bi-level formulation that is not present in prior HRL work. The experimental results also suggest an improvement of around 40% in success rates on harder sparse-reward tasks, compared to state-of-the-art baselines that already use preferences and primitive-informed regularization.

3. Clarity: The authors provide pseudo-code for their framework and clearly describe the alternation between preference-trajectory collection and updates of the high-level DPO objective, low-level value function, and low-level SAC policy. The introduction also clearly states the two central issues (non-stationarity and infeasible subgoals).

4. Quality: The derivation from bi-level HRL to a constrained problem, then to a Lagrangian, and finally to a primitive-regularized DPO objective is detailed and internally consistent.

**Weaknesses:**

1. The bi-level formalization in this paper is in some sense rephrasing standard HRL coupling (optimal lower-level policy conditioned on higher-level subgoals) in the language of constrained optimization. The authors could consider adding more analysis of the resulting bi-level problem such as convergence guarantees, regret bounds, and when the relaxed constraint and approximate value function yield near-optimal behavior.

2. This paper mentions that one benefit of DPO over RLHF is computational simplicity and stability. This statement is not quantified in terms of training time, memory etc for DIPPER vs RLHF-based alternatives (such as PIPER or a simple RLHF high-level baseline). ALso, sample efficiency is mainly reflected in success-rate vs timestep plots. There could be a more fine-grained analysis of preference-query complexity or number of environment transitions until convergence.

3. The paper claims their framework works on “long-horizon complex robotic tasks”, however the four environments are standard MuJoCo-style navigation/manipulation tasks (with some challenge enhancements like random mazes and sparse kitchen rewards). The work could consider adding vision-based setting with more complex hierarchical structures. Prior works like CRISP/PEAR do report real-robot experiments with more challenging perception and dynamics.

**Questions:**

1. Are preferences generated from ground-truth reward (e.g., via Bradley–Terry over cumulative environment reward), or from some kind of synthetic labeling procedure? How many preference queries are used per environment, and how does performance scale with the number of preferences?

2. The work approximates V^{L*} by updating V^L_m for m gradient steps between policy updates. How sensitive is DIPPER to m? Intuitively, if V^L_m is poor in the earlier portion of training, the regularizer might be actively pushing the high-level toward bad subgoals. Is there any safeguard (e.g., annealing λ) or empirical evidence that this does not happen?

3. Some suggestions on the presentation:
The work could benefit from having a more clear overview diagram that explicitly shows the bi-level view and how it leads to the primitive-regularized DPO block, with arrows from lower-level value function to the high-level DPO loss.
In the experiments, it would help to explicitly list the baselines and their key differences in a table, including whether they use preferences, whether they use primitive-informed regularization, and whether they are hierarchical or flat.

---

> ### Author Response · Authors · 2025-11-21
> **Response to Reviewer W15h [Part 1]**
>
> We would like to express our gratitude to the reviewer for dedicating their valuable time and effort towards evaluating our manuscript, which has allowed us to strengthen the manuscript. We deeply appreciate the insightful feedback provided, and we have thoroughly responded to reviewer's inquiries in the responses provided below.
>
> ## Weaknesses
>
> > **Weakness 1:** The bi-level formalization in this paper is in some sense rephrasing standard HRL coupling ... and approximate value function yield near-optimal behavior.
>
> **Response:** We thank the reviewer for this insight and agree that framing the HRL coupling as a constrained optimization problem opens avenues for richer theoretical insights which would better characterize when the relaxed constraints and approximate value functions lead to near-optimal policies.
>
> **Relaxed Constraint Formulation:** Considering the Equation 5 in the paper, we can re-formulate it as a *relaxed* constraint optimization problem as follows:
>
> \begin{split}
>     \max_{\pi^{H},\pi^{L}} \mathcal{J}(\pi^{H} , \pi^{L}) \hspace{0.3cm} s.t. \hspace{0.3cm} V^{L}(\pi^H) -  V^{L}_{*}(\pi^H) \geq \alpha,
> \end{split}
>
> where $\alpha > 0$ controls the slack for near-optimality. That said, our current work focuses primarily on the empirical validation of the proposed bi-level formulation of DIPPER using prefernce based learning in sparse-reward robotics tasks. We plan to extend this work by analyzing formal convergence or regret bounds in follow-up work.
>
> ---
>
> > **Weakness 2:** This paper mentions that one benefit of DPO over RLHF ... of preference-query complexity or number of environment transitions until convergence.
>
> **Response:** Please refer to our detailed response below:
>
> **Computational Simplicity of DPO in DIPPER:** DIPPER leverages DPO for higher-level optimization, bypassing the explicit reward modeling and RL steps required in RLHF-based alternatives, resulting in faster training and lower memory usage across benchmarks. In our results, and on hardware with an Intel Core i7, 48GB RAM, and NVIDIA GTX 1080 GPU, we train DIPPER for 1.35M timesteps for maze (∼3-4 hours), 0.9M for pick-and-place (∼4.5 hours), 0.62M for push (∼3.5 hours), and 0.63M for Franka Kitchen (∼4 hours). In contrast, we found that simple RLHF higher-level baseline achieved < 10% accuracy even after 2M timesteps (around 5.5 hrs in maze, 10 hrs on pick and place and bin, and 15 hours of training on kitchen task), and requires larger memory due to additional reward model fitting.
>
> **Sample Efficiency Analysis:** We quantify the sample efficiency in success-rate vs. timestep plots (Fig. 2), where we found that DIPPER reaches convergence faster when compared to PIPER (80-90% success in franka kitchen at ~0.62M steps, vs. PIPER which achieves <20% even at 2M timesteps). Similary, DIPPER reaches 35% success rate in pick and place and push tasks at ~0.8M steps, whereas PIPPER performance was <20% even after training for 2M timesteps. We will add this analysis in the final manuscript.
>
> ---
>
> > **Weakness 3:** The paper claims their framework works on “long-horizon complex robotic tasks”, ... like CRISP/PEAR do report real-robot experiments with more challenging perception and dynamics.
>
> **Response:** Our current experiments on standard MuJoCo benchmarks, which are particularly challenging due to sparse rewards (e.g., the agents are rewarded only when they solve the task, leading to exploration bottlenecks). While we focussed primarily on the empirical validation of our DPO based bi-level formulation in this paper, we thank the reviewer for the suggestion and agree that vision-based extensions (e.g., integrating VLMs for subgoal proposals) or real robot experiments would further enhance generalizability and credibility, and we plan to conduct sim-to-real transfer and real robot experiments in future work.
>
> ---

---

> ### Author Response · Authors · 2025-11-21
> **Response to Reviewer W15h [Part 2]**
>
> > **Question 1:** Are preferences generated from ground-truth reward (e.g., via Bradley–Terry over cumulative environment reward), or from some kind of synthetic labeling procedure? How many preference queries are used per environment, and how does performance scale with the number of preferences?
>
> **Response:** We address these concerns as follows:
>
> **Preference Dataset Collection Approach:** Our experiments utilized synthetic preferences derived from ground-truth environmental rewards to simulate human judgments. Specifically, we employed the standard Bradley-Terry model [1] where preferences are assigned based on cumulative rewards: P(prefer A over B) = $\frac{1}{1 + \exp(r_B - r_A)}$, where $r_A$ and $r_B$ are the total rewards for sequences A and B. Similar proxies have been utilized in the existing literature for preference-based RL algorithms [1,2,3] as it provides a controlled, noise-free environment to validate the algorithmic contributions (specifically the bi-level formulation and value regularization) before introducing the variability of human subjects.
>
> **Preference Dataset Details:** Since we use synthetic labeling procedure for collecting preferences, we leverage more than 10000 preference queries per environment (for the results in Figure 2). We found empirically that the performance does scale with the preference queries upto a point and then plateaus. We have provided the sensitivity analysis regarding robustness to noisy preference feedback in Table 2, and will add the analysis comparing the performance vs number of preferences in the final manuscript.
>
> [1] Christiano et al. "Deep reinforcement learning from human preferences" arXiv preprint arXiv:1706.03741, 2017.
>
> ---
>
> > **Question 2:** The work approximates V^{L*} by updating V^L_m for m gradient steps between policy updates. ... safeguard (e.g., annealing λ) or empirical evidence that this does not happen?
>
> **Response:** We address the sensitivity of DIPPER to the number of gradient steps m between policy updates below.
>
> **Sensitivity of DIPPER to $m$.** We would like to clarify that our experiments demonstrate that DIPPER remains robust to even low values of $m$. Specifically, while increasing $m$ improves success rates by approaching a near-optimal lower-level value function, we found that values as low as $m=2$ or $m=3$ achieve reasonably high performance. Thus, to balance computational efficiency with value estimation quality, we use small m (2 or 3) in our experiments.
>
> **Safeguards and Empirical Evidence.** Empirically, even low $m$ values of 2 or 3 do not lead to the generation of bad subgoals, as evidenced by the high success rates reported in our results. Consequently, we did not need to implement explicit safeguards such as annealing the regularization weight $\lambda$. For future work, we plan to explore conservative Q-learning [2] to mitigate risks from out-of-distribution subgoal states, ensuring the value function avoids overvaluing incorrect predictions.
>
>
> [2] Kumar et al. "Conservative Q-Learning for Offline Reinforcement Learning
> " arXiv preprint arXiv:/2006.04779, 2020.
>
> ---
>
> > **Question 3:** Some suggestions on the presentation: The work could benefit ... primitive-informed regularization, and whether they are hierarchical or flat.
>
> **Response:** We thank the reviewer and based on the reviewer's kind suggestions, we have replaced Figure 1 to better clarify the bi-level view of DIPPER in the revised manuscript.
>
> **Key Baseline Differences:** We provide a table clearly mentioning the key differences. We will add this in the final manuscript.
>
> | Method       | Uses Preferences | Primitive-Informed Regularization | Hierarchical/Flat |
> |--------------|------------------|-----------------------------------|-------------------|
> | DIPPER      | Yes (DPO)       | Yes (bi-level, value function-based)               | Hierarchical     |
> | DIPPER-No-V | Yes (DPO)       | No                               | Hierarchical     |
> | PIPER       | Yes (RLHF)      | No         | Hierarchical     |
> | SAGA        | No              | No          | Hierarchical     |
> | RAPS        | No              | Partial (behavior priors)        | Hierarchical     |
> | HAC         | No              | No                               | Hierarchical     |
> | HIER        | No              | No                               | Hierarchical     |
> | DPO-FLAT    | Yes (DPO)       | No                               | Flat             |
> | DAC         | No | No             | Flat             |
> | FLAT        | No              | No                               | Flat             |
>
> ---
>
> We hope these responses clarify the concerns. Please let us know if you need further clarifications.

---

### Author Response · Authors · 2025-12-03
**Rebuttal Discussion Summary for New Area Chair**

We thank all reviewers for their detailed and constructive feedback. During the rebuttal period, we recognized several important concerns and made substantial revisions to address them:

1. **Training Time Analysis Against RLHF Baselines (Reviewer W15h)**

   The reviewer asked for concrete training time comparisons between DIPPER and RLHF baselines. We added detailed computational metrics showing DIPPER's efficiency across environments. DIPPER requires the following training time: Maze ($\sim$ 3-4 hours, 1.35M steps), Pick-and-place ($\sim$ 4.5 hours, 0.9M steps), Push ($\sim$ 3.5 hours, 0.62M steps), Franka Kitchen ($\sim$ 4 hours, 0.63M steps). In contrast, the RLHF baseline achieves $<10\%$ accuracy even after 2M steps, requiring larger memory due to additional reward model fitting overhead.

2. **Low Performance vs Baselines in Simple Maze Task (Reviewer fdjG)**

   The reviewer questioned why some baselines (RPS, HAC, SAGA) outperform DIPPER in simple maze task. We clarified that DIPPER achieves $\sim$35% success in simple maze task while optimizing for scalability to complex tasks. The baselines outperforming here (RAPS, HAC, SAGA) rely on privileged information: RAPS uses pre-constructed behavior priors, HAC simulates optimal lower-level primitives, and SAGA uses state-conditioned discriminators which become harder to train as task complexity increases. We show empirically that these assumptions scale poorly to complex manipulation tasks (pick and place, push, Franka Kitchen) where DIPPER excels without requiring such domain-specific engineering.

3. **Expanded Related Work Coverage (Reviewers DfyA, fdjG)**

   To address the concerns, we expanded the related work section (Appendix A.3) to incorporate automatic curriculum learning approaches using bi-level optimization, intrinsic options for mitigating non-stationarity, and unified hierarchical optimization methods, thus positioning DIPPER more clearly within the broader landscape.

4. **Hyperparameter Robustness and Sensitivity (Reviewers W15h)**

   The reviewer raised questions about sensitivity of DIPPER to $m$ gradient steps when computing lower value function $V^L_m$. We clarified that DIPPER remains robust with low gradient steps ($m=2$ or $3$) while achieving high performance without requiring explicit safeguards like annealing. We also included a noise sensitivity analysis (Table 2) demonstrating robustness under noisy preference feedback.

5. **Clarification on Preference Data Source (Reviewer fdjG)**

   The reviewer raised concern regarding the paper's initial phrasing that inadvertently created ambiguity about whether human subjects were involved in preference annotation.  In the current work, our goal is to study the algorithmic and structural aspects of hierarchical preference-based RL under a controlled setting. To this end, **all experiments use synthetic preferences generated from ground-truth environment rewards via the Bradley–Terry model; no human subjects are involved**. We have revised Sections 1 and 4, as well as the Ethics Statement, to clearly describe this setup and remove any wording that could be read as implying human annotation.

   Reviewer fdjG initially interpreted the paper as using real human feedback and considered that a key empirical strength; after our clarification that the preferences are synthetic, reviewer lowered their score, emphasizing the importance of at least one human-in-the-loop experiment. We fully respect this perspective and agree that human (or VLM-based) feedback is an important evaluation axis. At the same time (as noted by Reviewers W15h and ywxX), we believe this axis is largely orthogonal to our main contributions: the bi-level HRL formulation, value-regularized DPO objective to mitigate non-stationarity and infeasible subgoal generation issues in HRL, and novel metrics to measure non-stationarity / subgoal-feasibility in HRL, which apply equally when preferences come from humans, VLMs, or a ground-truth reward proxy.

    Evaluating DIPPER with real human raters is therefore a natural next step, and we note this paper as establishing the core algorithmic framework and controlled empirical evidence on which such follow-up studies can build.

6. **Presentation and Visual Clarity (Reviewers W15h, DfyA)**

   We fixed all figure issues raised by reviewers; and improved the main overview diagram (Figure 1); added a comprehensive table explicitly comparing all baselines across three dimensions: whether they (i) use preferences, (ii) use primitive-informed regularization, and (iii) are hierarchical or flat.

We hope these clarifications and corresponding revisions address reviewer's concerns. DIPPER's core contributions: bi-level DPO for mitigating HRL non-stationarity and value function regularization for feasible subgoal generation, are now validated with enhanced empirical analysis, expanded baselines, and comprehensive sensitivity studies.

---

### Meta-Review · Area_Chair_NLex · 2026-01-07

**Summary:**

This paper proposes to incorporate the DPO objective into solving hierarchical RL tasks. The key idea is that, by replacing the high-level reward function $r(s_t, g_t, g^{star})$, which depends on the low-level policy, with the Bradley-Terry model’s reward, the proposed algorithm gets an improved stability over changes in the low-level policy. I think it can be think of that as long as the changes in the low-level policy doesn’t alter the order of two rewards (say $r(s_1, g_1, g^{star})$ and $r(s_2, g_2, g^{star})$), the proposed objective remains “stable”.

**Reviewer Concerns:**

The reviewers have split opinions about the paper. On one hand, two reviewers think the formulation of the stability problem is valid and the proposed solution is an interesting step towards solving the problem. On the other hand, other reviewers raised valid concerns regarding the simplicity and fairness of the empirical evaluation. Specifically, the preference data is provided by the groundtruth reward function only (not labeled by humans), and more ablation studies are required to justify the improvements over the baselines (e.g., incorporating HER into DIPPER).

Overall, I think the paper points out an important problem and the proposed solution matches my intuition, and the merits of the paper outweigh its problems. Therefore, I recommend acceptance of the paper. However, I strongly encourage the authors to revise the paper and add necessary experiments according to the reviewers’ suggestions.

**Reviewer Scores:**

W15h: They will likely maintain their score as the rebuttal addressed the main concerns sufficiently.

ywxX: Given the low confidence of the reviewer and the main concerns not being very specific, I think the reviewer may not change the score.

DfyA: The reviewer liked the paper and might actually champion it.

fdjG: The reviewer mentioned that they may decrease the score due to the absence of experiments with actual human preference labels.

---

### Decision · Program_Chairs · 2026-01-26

Accept (Poster)